# Non-coding genetic variants underlying higher prostate cancer risk in men of African ancestry

Shan Li [1], Kaniz Fatema [2], Nidharshan Sundarraj[3], Arashdeep Singh [1], Padma Sheila Rajagopal [1], Dimple Notani [3], David Y. Takeda [2] & Sridhar Hannenhalli [1] ✉

Prostate cancer (PrCa) incidence and severity vary across ancestries; men of African ancestry (AA) are more likely to be diagnosed and die from PrCa than those of European ancestry (EA). Current polygenic risk scores, even from multi-ancestry GWAS, do not fully capture population-specific genetic mechanisms, especially those mediated by non-coding regulatory single nucleotide polymorphisms (SNPs). Using a deep learning model of prostate enhancers, we identify ~2000 SNPs, potentially affecting enhancer function, with higher alternate allele frequency in AA men, that may affect PrCa risk. These SNPs may promote cancer via two mechanisms: increased enhancer activity leading to immune suppression and telomere elongation or decreased activity causing de-differentiation and apoptosis inhibition. Identified SNPs predominantly modulate binding of key transcription factors such as FOX, HOX, and AR – the first was experimentally validated. Incorporating these SNPs into a polygenic risk score improves PrCa risk assessment beyond existing GWAS-identified variants.

PrCa exhibits considerable heritability and notable disparities in incidence rates among different ancestral populations, with men of AA displaying a far greater risk than those of European (1.75-fold) and Asian (3.18-fold) descent[1,2]. Additionally, AA men experience the highest mortality rate from PrCa[3]. Studies have demonstrated a role for population-specific genetic factors, even in conjunction with neighborhood deprivation and other social determinants of health, in PrCa incidence[4–6]. Large genome-wide association studies (GWAS) have identified PrCa risk variants, which are aggregated into a polygenic risk score (PRS)[7–10]. In general, GWAS-derived risk variants do not generalize well to independent discovery cohorts[11–13]. Specifically, PrCa PRSs, including those based on multi-ancestry cohorts, show variable accuracies across ancestries[14]. More importantly, variants discovered via GWAS do not directly point to underlying mechanisms.

Recent studies have shown that a rare, AA-specific variant of the non-coding SNP rs72725854 (A > G/T) at the 8q24 locus is associated with a ~2-fold increased risk of PrCa[15,16]. Follow-up work showed that the alternate allele "T" introduces a binding site for the transcription factor (TF) SPDEF, which substantially increases the activity of the enhancer harboring the SNP and, in response to androgen stimulation, increases the expression of the enhancer's target genes PCAT1, PVT1, and c-Myc in prostate tumors; enhancer with the reference "A" allele remain non-responsive to androgen[17]. This finding underscores the potential roles of germline non-coding regulatory variants in mediating cancer risk, motivating this study.

Here, we aim to comprehensively assess the role of genome-wide non-coding variants associated with higher PrCa risk in AA men, specifically via affecting regulatory enhancer function in prostate tissue. We propose a method that integrates a sequence-based deep learning

[1]Cancer Data Science Laboratory, Center for Cancer Research, National Cancer Institute, National Institutes of Health, Bethesda, MD, USA. [2]Genitourinary Malignancies Branch, Center for Cancer Research, National Cancer Institute, National Institutes of Health, Bethesda, MD, USA. [3]Genetics and Development, National Centre for Biological Sciences, Tata Institute of Fundamental Research, Bangalore, Karnataka, India. ✉e-mail: sridhar.hannenhalli@nih.gov

enhancer model with a sliding window strategy, referred to as ProSNP-DL. The deep learning model of enhancer activity has single-nucleotide resolution[18], i.e., the model can predict and quantify the potential change in the enhancer activity due to a single nucleotide change in the enhancer sequence. Applying ProSNP-DL genome-wide, we identify 2407 non-coding SNPs (termed enhancer SNPs or eSNPs) with a greater frequency of the alternate allele in AA individuals, that exhibited a substantial difference in their predicted enhancer activities between the two alleles. Of the 2407 eSNPs, for 1296 (53.8%) eSNPs, the alternate allele is predicted to have a higher activity (we refer to these as "gained" eSNPs), and for the remaining 1111 eSNPs, the alternate allele is predicted to have a lower enhancer activity (we refer to these as "lost" eSNPs). We noted complementary functions affected by the gained and the lost eSNPs: putative targets of gained eSNPs are involved in oncogenic pathways, such as suppression of the adaptive immune system and telomere elongation. In contrast, targets of lost eSNPs are involved in tumor suppression pathways, including differentiation. Gained and lost eSNPs respectively establish and inhibit the binding of several families of TFs, in particular those critical in prostate development and homeostasis, as well as PrCA development, including FOX, AR, and HOX. Finally, we assessed the ability of eSNPs to quantify PrCa risk. We found that a PRS based on eSNPs can distinguish PrCa patients from healthy controls with high accuracy, far better than controls. Importantly, eSNPs, which are biologically motivated and derived agnostic to PrCa status, add value to the previously reported gold-standard PRS learnt from a large multi-ancestry cohort of PrCa patients and healthy controls[8]. Overall, our biologically informed approach effectively identifies regulatory germline variants linked to elevated PrCa risk, while also suggesting potential mechanisms.

## Results

### Overview of the study

To identify the regulatory variants associated with divergent PrCa risk in AA and EA men, we first restricted our investigation to 10,171,946 non-coding SNPs that were common (Minor Allele Frequency; MAF > = 5%) in either the EA or the AA population, and further selected the 491 K SNPs having the most disparate allele frequencies between AA and EA (top 5% $F_{ST}$ score). We next prioritized these candidate SNPs in silico based on their effect on prostate enhancer activity as follows. Leveraging the H3K27ac ChIP-seq data (as a proxy for active enhancers) in a prostate-derived LNCaP cell line, we trained a deep learning model that learns the sequence encoding of prostate enhancers and is capable of quantifying the regulatory impact of a SNP. Here, the major and minor alleles in EA are considered the reference and the alternate alleles, respectively. Our overall approach involves a deep learning model of enhancer activity and a sliding window strategy, to account for all potential position of a SNP within an enhancer, aiming to identify SNPs potentially affecting enhancer activity (Fig. 1, Methods); we dubbed our approach as ProSNP-DL (Prostate Cancer Risk SNPs prediction using Deep Learning). Applying ProSNP-DL to the aforementioned 491 K SNPs, we identified two sets of AA-dominant SNPs (defined as those with alternate allele frequency higher in AA than EA) whose alternate alleles are predicted to lead to either an enhancer gain or loss, referred to as gained (1296) and lost (1111) eSNPs, respectively. Figure 1 illustrates our approach. The rationale underlying this approach is that the SNPs with large potential to affect enhancer activity (either gain or loss) are more likely to have a phenotypic impact in the corresponding cellular context. Next, we investigated the biological pathways associated with the two sets of eSNPs and identified the potential TFs whose bindings are likely disrupted by the eSNPs. We further conducted a ChIP-seq experiment targeting FOXA1, a primary cognate TF of the eSNPs, in two prostate cancer cell lines (LNCaP[19], derived from a donor of European ancestry and MDA PCa 2B[20], derived from a donor of African ancestry) to assess the functional impact of eSNPs on TF binding disruption. Finally, we derive a PRS

from the eSNPs and assess its potential to quantify PrCa risk, relative to the PRSs based on previously published risk variants derived from GWAS.

### ProSNP-DL - a deep learning model of PrCa enhancers with single-nucleotide resolution

To capture the sequence encryption of the PrCa enhancers, we first trained a deep convolution model (Fig. 2A, Methods) on 67,064 LNCaP H3K27ac peaks[21] as a proxy for prostate enhancers, and 10-fold pan-cell type DNase Hypersensitive regions not overlapping LNCaP H3K27ac peaks as the negative control to ensure the tissue-specificity of the PrCa enhancer model. The enhancer model has the same architecture as the one in our previous work[18]. The model is able to accurately discriminate prostate enhancers from accessible regions devoid of prostate enhancers, with the area under the receiver operating characteristic curve (auROC) of 0.91 (Fig. 2B). We have shown that an enhancer model in HepG2 cell line with the same architecture and parameters is able to predict the allele-specific effects of previously profiled reporter assay QTLs (raQTLs)[22] on enhancer activity[18]. In short, the enhancer model score is not only able to identify active enhancers, is sensitive to single nucleotide changes in enhancer activity. Notably, we applied the LNCaP enhancer model to prioritize non-coding genetic variants after ascertaining that the enhancer sequence features of LNCaP cells closely resemble those of other prostate cancer cell lines and normal primary prostate tissues (Fig. S1 and supplementary information section I).

However, the effects of a SNP on enhancer activity depends on the position of the enhancer (Fig. 2C), as enhancers overlapping the SNP at various locations have different sequence contexts. We therefore evaluate a SNP's influence by aggregating its effects across all conceivable positions of a hypothetical enhancer within a 1 KB window. The number of windows in which the alternate allele is predicted to substantially transform a neutral sequence into an active enhancer or vice versa is termed the "essential window number" (EWN). We utilized the EWN to quantify the effect size of the SNP on enhancer activity (Methods), as we expect a more consistent directional impact of functional SNPs on enhancer activity in a larger sequence context. Specifically, if the alternate allele consistently exhibits positive delta scores (the difference in model scores caused by the alternate allele) and converts a neutral sequence to an active enhancer in at least 5 windows (EWN ≥ 5), the SNP is classified as a gained activity eSNP (gained eSNP for simplicity), suggesting an enhancer activity gain. Conversely, if the alternate allele consistently shows negative delta scores and alters an active enhancer to a neutral sequence in at least 5 windows (EWN ≥ 5), the SNP is considered a lost activity eSNP (lost eSNP for simplicity; Fig. 1, Methods), suggesting an enhancer activity loss. Figure 2C illustrates this logic for gained eSNPs using the aforementioned PrCa risk SNP rs72725854 that were previously identified[17]. The delta scores of alternate allele "T" are larger than the reference allele "A" in all the windows in the (−200 bp, 200 bp) region centered by the SNP, and change an inactive enhancer to an active enhancer (with False Positive Rate FPR < 0.01) in 5 windows (EWN = 5) according to our model. In summary, our approach aims to identify SNPs with a significant impact on enhancer activity by integrating a deep learning model trained on context-specific enhancer sequences with a sliding-window strategy, termed ProSNP-DL.

### ProSNP-DL identifies enhancer SNPs dominant in an African ancestral population and associated with PrCa risk and patient survival

Starting with 84,802,133 genome-wide SNPs, we first selected the 10,171,946 non-coding SNPs that are common (MAF > = 5%) in EA or AA, to ensure the robustness of our analysis. Further, to specifically identify SNPs that may underlie the higher PrCa incidence in AA men, we focused on the SNPs with the largest allele frequency difference

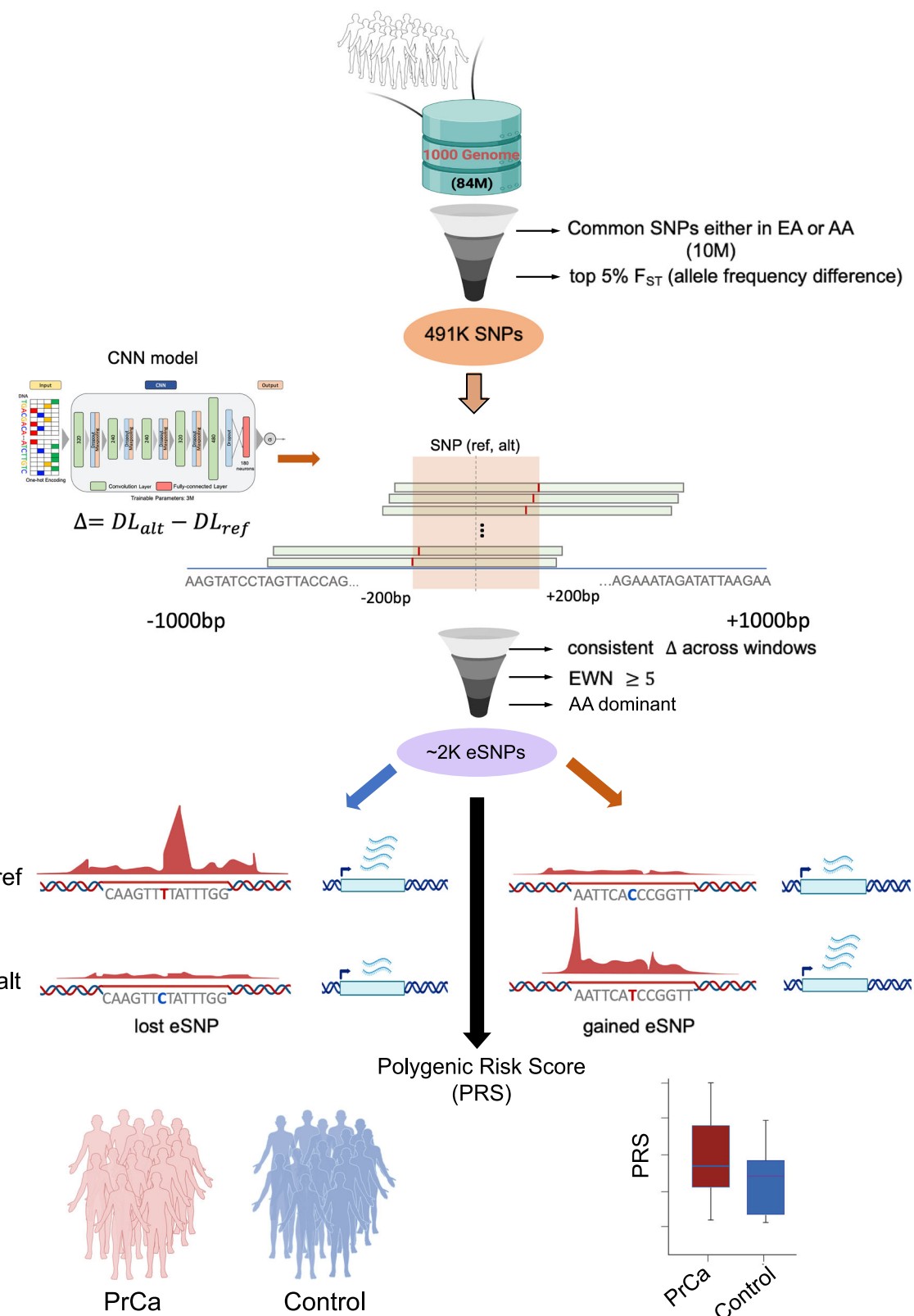

**Fig. 1 | The flow chart of our pipeline.** Non-coding SNPs in 1000 Genomes Project are retained if they are common in either EA or AA cohort, and have a high allele frequency difference between the two populations. The deep learning model integrated with a sliding window strategy was then applied to the retained 491 K SNPs to select the ~2000 AA-dominant SNPs (eSNPs) at which the model predicts substantially different enhancer activities for the two alleles, using a criterion called essential window number (EWN) that measures consistency of the directional impact of functional SNPs on enhancer activity (Methods). The eSNPs are categorized as either gained, where the alternate allele has greater predicted enhancer activity, or lost, where the alternate allele has lower predicted enhancer activity. These ~2000 eSNPs are used to assess the polygenic risk score (PRS) for PrCa for an individual. Part of the figures are generated by BioRender.

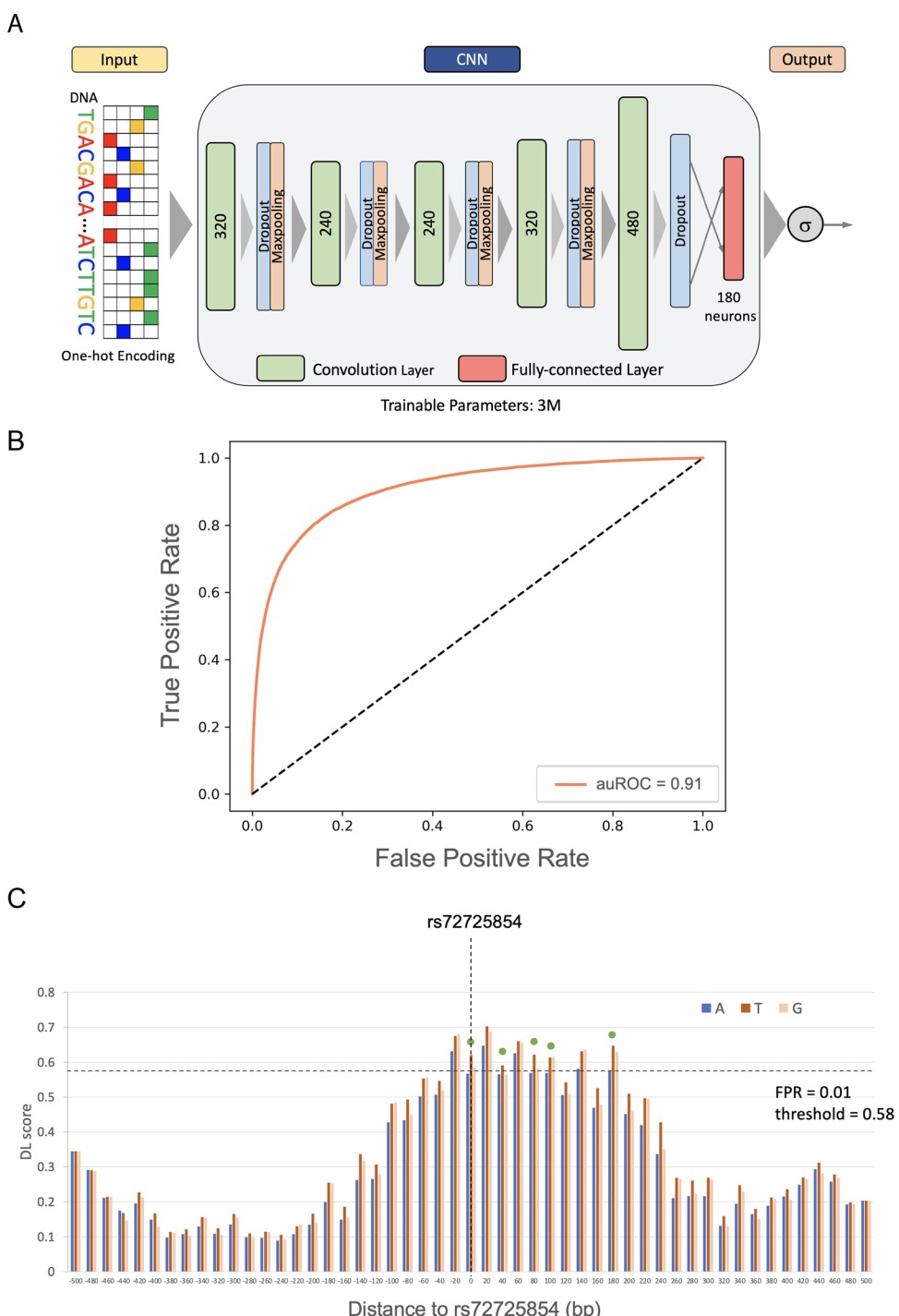

**Fig. 2 | Architecture and performance of enhancer model. A** The architecture of the deep convolutional neural network. **B** ROC curve of the model in predicting PrCa enhancers. **C** Deep learning model-predicted scores (DL scores) of the reference and alternate alleles of the previously reported PrCa risk SNP rs72725854[17] across the sliding 1KB windows overlapping the SNP rs72725854. The x-axis is the relative distance of the window center to the SNP. The essential positions are highlighted by a green dot. Source data for these figures are provided as a Source Data file.

between AA and EA (Methods), yielding 491 K SNPs (Fig. S2A). Applying ProSNP-DL on these SNPs yielded 2069 gained enhancer activity SNPs, and 2175 lost enhancer activity SNPs (eSNPs). Overall, these included 2407 eSNPs with higher alternate allele frequency in AA (AA-dominant), and 1837 eSNPs with higher alternate allele frequency in EA (EA-dominant) (Fig. S2B). The remainder of the 491 K SNPs are referred to as non-enhancer SNPs (non-eSNPs) and used as a negative control for several following analyses. A priori, we are equally interested in both AA-dominant and EA-dominant e-SNPs, as they could both reveal

major functional differences between EA and AA associated with PrCa. To validate the functional impact of eSNPs on enhancer activity, we assessed allelic imbalance of H3K27ac reads at the heterozygous eSNP sites. Compared with non-eSNPs, the eSNPs exhibited a significantly greater extent of allelic imbalance of H3K27ac read coverage; as expected, the alternate alleles at gained eSNPs showed larger H3K27ac read coverage (Fig. 3A), and those at lost eSNP exhibited lower H3K27ac read coverage (Fig. 3B). This result suggests a coherent functional impact of the identified gained and lost activity eSNPs on

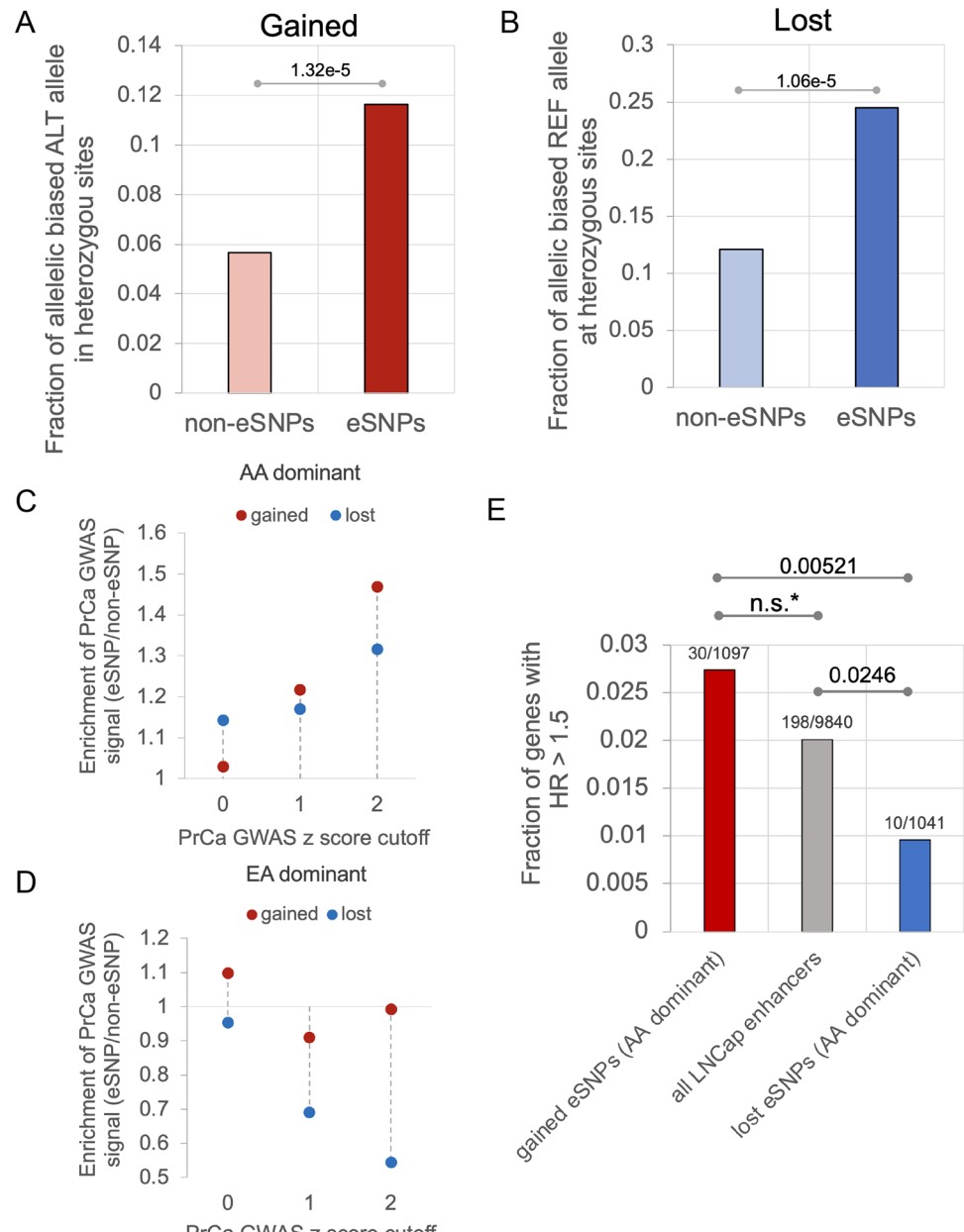

**Fig. 3 | eSNPs with AA dominant alternate alleles are associated with PrCa risk.**
Fraction of heterozygous eSNPs (**A** Gained, **B** Lost) that exhibit allelic imbalance. *P*-values were derived from one-sided Fisher's exact test. **C** Enrichment of both AA-dominant gained and lost activity eSNPs in PrCa GWAS traits compared to non-eSNPs. An increased enrichment pattern of both gained and lost AA-dominant eSNPs in the PrCa GWAS signal across increasing cutoffs can be observed.
**D** Enrichment of both EA-dominant gained and lost activity eSNPs in PrCa GWAS traits compared to non-eSNPs. In **C**, **D** The enrichment (*y*-axis) was measured as the ratio of (i) the fraction of eSNPs with PrCa GWAS signal *z*-score greater than the

signal cutoff (*x*-axis) to (ii) the fraction of non-eSNPs with PrCa GWAS signal *z* score greater than the cutoff (*x*-axis). **E** In the three categories—AA-dominant gained activity eSNPs, lost activity eSNPs, and all LNCaP enhancers, the figure shows the fraction of genes with HR > 1.5 (Bonferroni-corrected *P*-value ≤ 0.05). Each pair of categories was compared using one-sided Fisher's exact test. n.s. refers to non-significant based on Fisher's exact test. * refers to significance using bootstrap-based approach: *p*-value = 0.0189 (Methods). Source data for these figures is provided as a Source data file.

the experimentally determined gain and loss of enhancer activity, respectively.

Next, we directly compared the AA-dominant and EA-dominant eSNPs in terms of their association with PrCa risk, using two complementary strategies. First, we assessed the association of the eSNPs with PrCa risk, using summary statistics (z-score normalized GWAS signal) from a PrCa GWAS of 140,306 males[10]. Notably, the eSNPs with a greater alternate allele frequency in AA are strongly associated with the PrCa GWAS traits (Fig. 3C). Specifically, overall, there is an increased enrichment pattern of both gained and lost AA-dominant

eSNPs in the PrCa GWAS signal, and this enrichment becomes stronger as the GWAS signal cutoff (*x*-axis) becomes more stringent (Fig. 3C and Fig. S2C); Contrastingly, this is not the case for the eSNPs with EA-dominant alternate allele frequency (Fig. 3D, Fig. S2C). As our second strategy, for the AA-dominant gained activity eSNPs, we assessed whether their putative target genes (proximal genes taken as the proxy) are associated with worse patient survival using Cox regression, controlling for age; conversely, for the lost activity eSNPs, we assessed whether their putative target genes are associated with better survival. Consistent with the first strategy, the target genes of the gained

activity AA-dominant eSNPs associated with decreased survival, as shown by the high hazard ratios (HR), and those of the lost AA-dominant eSNPs with improved survival, as shown by low HRs (Fig. 3E). However, these associations were not observed for the EA-dominant gained and lost activity eSNPs (Fig. S2D). Taken together, our findings are indicative of a distinct association between eSNPs and PrCa risk and survival for AA-dominant eSNPs encoding either gained or lost enhancer activity. In the following, we focus specifically on the AA-dominant gained (1296) and lost (1111) eSNPs and further investigate their role in prostate cancer risk.

### The gained and lost activity eSNPs may facilitate carcinogenesis via complementary mechanisms

We next functionally characterize gained and lost activity eSNPs, using the annotations enrichment tool GREAT[23,24] (Method). The analysis showed that gained activity eSNPs are notably associated with immune response and regulation of the immune system, and telomerase activity regulation (Fig. 4A). This suggests that the gained activity eSNPs may facilitate carcinogenesis by modulating the immune system, substantiated by prior clinical research[25]. To further assess this possibility, we identified the subset of gained activity eSNPs, whose alternate alleles were present in at least ten ICGC PrCa samples. We then compared the T cell dysfunction scores from TIDE (Tumor Immune Dysfunction and Exclusion)[26] of the genes proximal to such eSNPs with genes near other gained activity eSNPs and non-eSNPs. We observed that the genes near the frequently gained activity eSNPs exhibit higher T cell dysfunction scores (Fig. 4C, Methods), suggesting potential functional impact of the gained activity eSNPs on immune system dysfunction. In addition, the gained activity eSNPs are associated with the regulation of telomerase activity. Several lines of evidence indicate that longer telomere length contributes to cancer[27,28]. In agreement with these findings, a previous study has shown that relative to cells derived from Europeans, the cells derived from Africans have longer telomere length across all tissues and, even more relevant, the difference was found to be the highest in the prostate[29], further supporting a potential causal role of the gained eSNPs in tumor progression.

In contrast, the lost activity eSNPs are largely associated with the regulation of differentiation/development (Fig. 4B). We hypothesized that genes regulated by lost activity eSNPs are involved in maintaining differentiated states, and the loss of their enhancers may cause cells to revert to a more undifferentiated state. In other words, the target genes of lost activity eSNPs are expected to be highly expressed in adult normal prostate tissue but repressed in the developmental stage. To validate this hypothesis, we performed a regression analysis to assess the association of different classes of SNPs to the adult-to-fetal prostate differential expression. As shown in Fig. 4D, a positive coefficient for the target genes of lost activity eSNP indicates their inactivation during development, and conversely, the target genes of gained activity eSNPs exhibit higher expression in the fetal prostate compared to the adult normal prostate (GTEx), consistent with our hypothesis (Fig. 4D). Notably, when we stratify the eSNPs based on the number of patients in which the risk allele occurs, the more frequent risk alleles show a greater magnitude of coefficient (Fig. 4D), further supporting their functional impact. It is worth noting that the genes proximal to gained eSNPs are largely involved in suppression of the immune system (Fig. 4A, C), which is an essential feature of fetal development[30,31]. Therefore, it does not seem surprising that these genes are highly expressed in the fetal prostate compared to the adult prostate. Furthermore, the target gene of gained activity eSNPs have higher Cancer Dependency score (DepMap, Methods) compared to those of lost activity eSNPs (Fig. 4E), suggesting their oncogenic role. To further investigate the relevance of eSNP-associated genes to tumor progression—specifically their pro- or anti-tumorigenic potential—we evaluated whether genes linked to gained eSNPs, particularly those

enriched in immune response, regulation of immune system, and regulation of telomere maintenance Gene Ontology (GO) terms, are associated with poorer patient survival, and whether genes linked to lost eSNPs, enriched in developmental and differentiation-related GO terms, correlate with improved survival outcomes. Notably, genes within the GO terms associated with gained eSNPs consistently exhibited elevated (HR > 1), indicative of pro-tumorigenic potential, whereas those associated with lost eSNPs consistently showed reduced (HR < 1) (Fig. S3, Supplementary Data 1–2), suggestive of tumor-suppressive functions. Together, these findings suggest that gained and lost eSNPs may collaboratively promote carcinogenesis through complementary mechanisms—by activating oncogenic pathways and impairing tumor suppressive functions.

### eSNPs modulate binding of key prostate transcription factors including the FOX and the HOX families

The identified eSNPs, by design and supported by further validations, affect enhancer activity, which might be mediated, in substantive part, via disruption (gain or loss) of TF binding[17,32–35]. We therefore sought to identify the cognate TFs whose bindings are potentially disrupted by the eSNPs. Toward this, for both gained and lost activity eSNPs, we focused on the TFs whose binding sites are enriched at the eSNPs sites compared to the non-eSNP sites. This analysis was conducted using sequences containing both the reference and alternate alleles to eliminate allelic bias (Methods). Our findings revealed that for both gained and lost activity eSNPs, TFs belonging to the FOX and HOX families were among the most enriched (Supplementary Data 3–6). The eSNPs affecting these TFs not only exhibited large effects on enhancer activity (average EWN ≥ 7) but also had the highest number of potential target genes (Fig. 5A, B). This finding aligns with previous research indicating that FOXA1 acts as a pioneer factor in PrCa, capable of extensively reprogramming the androgen receptor (AR) cistrome in collaboration with HOXB13[36]; interestingly, AR is also among the TFs majorly disrupted by gained activity eSNPs (Fig. 5A, B, Supplementary Data 3–4). This provides further mechanistic insights into how eSNPs might contribute to PrCa risk. Although gained and lost activity eSNPs appear to disrupt the binding of the same set of TFs, we speculated that their target genes are involved in complementary pathways. Indeed, the genes proximal to gained activity eSNPs associated with FOX family TFs were enriched in the cancer signatures, such as hypoxia, epithelial-to-mesenchymal transition (EMT), metastasis, and stemness signatures in the CancerSEA database[37]. In contrast, the genes proximal to lost eSNPs associated with FOX were enriched in the cancer signatures, including differentiation and apoptosis (Fig. 5C, Methods). To further examine the association between eSNP-linked genes and tumor progression—specifically their potential roles in promoting or suppressing tumorigenesis—we analyzed whether genes associated with gained eSNPs, particularly those enriched in hypoxia, EMT, metastasis, and stemness-related CancerSEA signatures, are linked to poorer patient survival, and conversely, whether genes associated with lost eSNPs, enriched in differentiation and apoptosis signatures, are correlated with improved prognosis. As hypothesized, gene sets related to gained eSNPs consistently exhibited increased hazard ratios (HR > 1), reflecting pro-tumorigenic effects, while those related to lost eSNPs exhibited decreased hazard ratios (HR < 1) (Fig. S4, Supplementary Data 7, 8), indicating tumor-suppressive functions.

To experimentally validate the impact of eSNPs on FOX binding, we performed ChIP-seq for FOXA1 in two PrCa cell lines of distinct ancestral origins: LNCaP cells, derived from a lymph node metastasis of a 50-year-old Caucasian male with prostate adenocarcinoma[19], and MDA PCa 2B cells, isolated from the prostate of a 63-year-old Black male patient with adenocarcinoma[20] (Methods). First, consistent with our expectation, MDA PCa 2B has an overall greater number of alternate alleles at eSNP sites, aligning with the fact that these eSNPs are AA

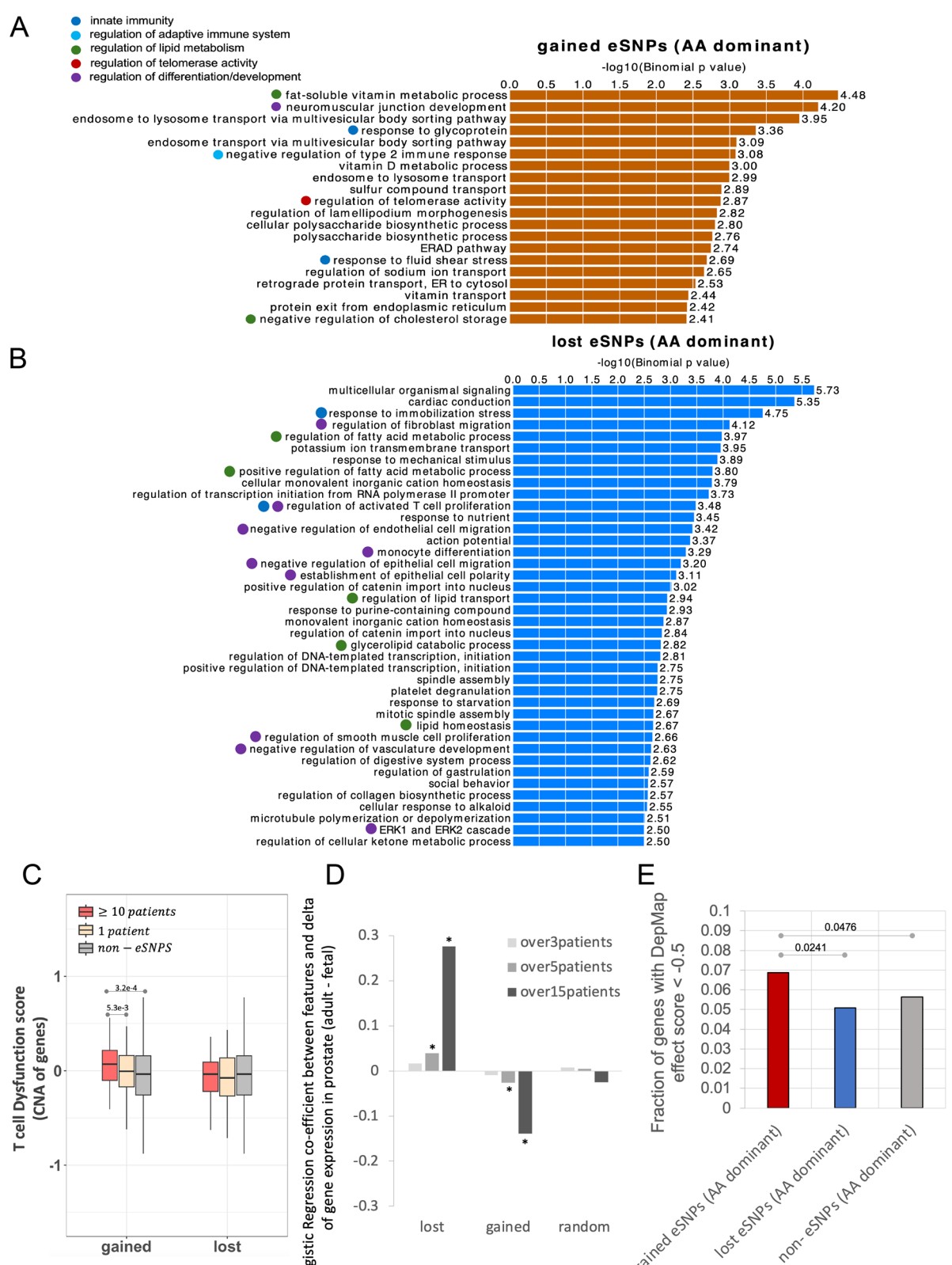

dominant (Supplementary Data 9). Furthermore, we observed that relative to LNCaP, FOXA1 binding intensity was greater in MDA PCa 2B at gained activity eSNP sites and lower at lost activity eSNP sites (Fig. 5D, "Methods").

One lost activity eSNP coincided with a common SNP rs10095018 (T/C), which is spatially proximal to the gene *NDRG1* in the 3D genomic space (Fig. S5A), where the alternate allele "C" substantially reduces

enhancer activity according to our model (EWN = 10). NDRG1 is recognized for its role in suppression of PrCa; decreased membrane expression of *NDRG1* has been associated with significantly poorer survival outcomes in PrCa patients, and is correlated with higher Gleason scores, which predict the aggressiveness of PrCa[38,39]. At the rs10095018 SNP site, the LNCaP cell line possesses two reference alleles (T/T), while the MDA PCa 2B cell line has heterozygous alleles

**Fig. 4 | Gained and lost activity eSNPs with AA-dominant alternate alleles are associated with PrCa risk through two complementary mechanisms.** Biological processes associated with gained activity eSNPs **A** and with lost eSNPs **B** using the GREAT tool. The bar values are the -log10 of the P-values based on a one-sided Binomial test without multiple test adjustment. **C** Average T cell dysfunction score across cancer types of the genes near the gained and lost activity eSNPs, as well as non-eSNPs as a control. The genes near the frequently (occurred in at least 10 patients) gained activity eSNPs (n = 349) exhibit higher T cell dysfunction scores than the genes near the eSNPs occurred only in one patient (n = 292), and the genes near non-eSNPs (n = 10,000). P-values are obtained from a one-sided Wilcoxon test. In the boxplot, the horizontal line in the middle is the median value, and the lower and upper edges of the boxes correspond to the 25th and 75th percentiles. Extending vertically upwards/downwards of the boxes are the lines showing 1.5 times the interquartile range (i.e., distance between 25th and 75th percentile). **D** Coefficients for logistic regression of gene expression change (adult - fetal) against genes associated with various sets of SNPs. * indicates coefficient significance (two-sided Wald test p-value) <= 0.05. E). Fraction of different sets of genes with DepMap effect score < −0.5. One-sided Fisher's exact test was applied to evaluate significance. All p-values are Bonferroni-corrected. Only significant p-values are shown in the plots. Source data for these figures is provided as a Source data file.

(T/C). Notably, LNCaP exhibited a stronger FOXA1 binding signal compared to MDA PCa 2B (Fig. 5E). Furthermore, the MDA PCa 2B cell line exhibits allelic imbalance at this position, with the C allele harboring far fewer ChIP-seq reads of FOXA1 (Fig. S5C). These observations together suggest that the alternate allele C can indeed disrupt FOXA1 binding. More importantly, *NDRG1* expression level is higher in LNCaP compared to MDA PCa 2B (Fig. 5F), suggesting that the alternate allele "C" at lost eSNP rs10095018 reduces the expression of tumor suppressor NDRG1 by disrupting FOXA1 binding in MDA PCa 2B cell line, thus adversely affecting PrCa patient outcome. Next, going beyond NDRG1 locus, to further assess the overall impact of eSNPs on the expression of putative target genes, we compared gene expression differences between the MDA and LNCaP cell lines using expression data from DepMap, focusing on genes that are in 3D contact (via HiChIP loops) with either gained or lost eSNPs. Consistent with expectation, the MDA cell line—which harbors a greater number of alternate alleles at gained eSNPs compared to LNCaP—exhibited higher gene expression at these sites. Conversely, at lost eSNPs, the MDA cell line showed reduced expression relative to LNCaP (Fig. S6). In summary, these eSNPs may contribute to heightened PrCa risk by disturbing the binding of key TFs and thereby changing the expression of key genes.

### Polygenic risk score derived from eSNPs can be helpful to differentiate between cases and controls of PrCa

While by design our eSNPs have high allele frequency differential between AA and EA ancestry, given their potential functional role in prostate cancer, here we assessed their ability to broadly capture PrCa susceptibility across various ancestral populations. We derived a PRS based on the number of risk alleles of each eSNP in an individual, trained and tested (10-fold cross-validation) in a large EA cohort in a PrCa GWAS study (PEGASUS) (4599 cases and 2841 controls, see Methods). We compared the cross-validation accuracy of eSNP-based PRS with those based on (i) randomly select 5000 high Fst SNPs, (ii) 50,708 chromatin Quantitative Trait Loci (cQTLs) linked to chromatin accessibility in PrCa[7] and (iii) 545 PrCa risk variants (PGS-545) identified through a multi-ancestry genome-wide study[8]. We followed the convention[8] to measure the accuracy as follows: we partitioned the range of PRS scores for the controls into five quantiles and assessed the enrichment of PrCa PRS scores in the top and the bottom quantiles. As shown in Fig. 6A (and Fig. S7), eSNP-based PRS substantially outperformed those based on cQTL and random high $F_{ST}$ SNPs. As expected, PGS-545 has a superior performance, as reported previously[8]. However, integrating eSNPs with PGS-545 SNPs resulted in a slightly improved performance (Figs. 6B and S3), suggesting that eSNPs provide complementary information to PGS-545 SNPs. These results observed in the cross-validation within the EA cohort were similar to those when a model trained on the EA cohort was applied to the independent AA cohort (474 cases and 458 controls; see Methods). These results suggest that eSNPs provide valuable insights into PrCa susceptibility across various ancestral populations. Interestingly, we also found that, among the AA PrCa patients, those with higher PRS based on eSNPs and PGS-545 tend to develop PrCa at younger ages compared to those with lower PRS (Fig. 6C), supporting the role of eSNPs and PGS-545 SNPs in PrCa onset (see Methods).

Given that eSNPs are identified agnostic of PrCa cohorts, it is not surprising that they do not perform as well as the gold-standard GWAS-based PGS-545 SNPs in discriminating PrCa from controls. However, eSNPs are more likely to be functional in terms of affecting enhancer function, and they are likely to underlie prostate-related disorders other than PrCa. First, we found that compared to PGS-545 SNPs, eSNPs indeed exhibit a greater allelic imbalance for H3K27Ac reads (Fig. 6D). Looking at the GWAS catalog, while there are four additional traits besides PrCa related to the prostate, they are all indirectly related to cancer—prostate antigen levels and various adverse responses to PrCa treatment. We assessed whether eSNPs better capture these additional traits than PGS-545, which is based specifically on PrCa. We found that eSNPs indeed show a greater enrichment for these additional traits (Fig. 6E). Furthermore, the enrichment of these prostate traits in eSNPs is true only for AA-dominant eSNPs; there is no overlap between EA-dominant SNPs and the QTLs for these additional traits. These results suggest that by virtue of being mechanism-based and agnostic to PrCa, eSNPs may better capture other prostate-related traits.

Overall, these results suggest that our identified eSNPs, derived in mechanism-aware but PrCa-agnostic fashion, can contribute to prioritize PrCa susceptibility across different ancestries, as well as other prostate disorders.

### Discussion

In this study, we explored non-coding regulatory polymorphisms potentially underlying the high prevalence of PrCa and associated mortality in AA men. Using a sequence-based deep learning model (ProSNP-DL) that learns the sequence encodings of prostate enhancers and is capable of quantifying regulatory effects of a single nucleotide change, we identified two sets of regulatory genetic variants: one included the SNPs where the alternate allele encoded an increased enhancer activity (gained activity eSNPs) and the other included the SNPs where the alternate allele encoded a reduced enhancer activity (lost activity eSNPs). We particularly focused on the eSNPs with greater alternate allele frequency in AA (AA-dominant SNPs) because: (1) the AA-dominant eSNPs tended to have a stronger signal for PrCa GWAS traits than EA-dominant eSNPs, and (2) a higher expression of the target genes of AA-dominant eSNPs was commonly associated with patient survival, which was not observed for EA-dominant eSNPs. Overall, we identify ~2000 gained and lost activity eSNPs with AA-dominant alternate alleles that provide a list of regulatory variants potentially associated with the excessive PrCa risk manifest in AA men.

We discovered that the gained and lost activity eSNPs potentially affect PrCa risk via two complementary mechanisms. Specifically, the genes near the gained activity eSNPs exhibit a higher T cell Dysfunction score, leading to T cell exhaustion and suppression of the adaptive immune system. Moreover, the gained activity eSNPs are linked to telomere elongation, a process implicated in oncogenesis[27,28]. On the other hand, the lost activity eSNPs are associated with dedifferentiation (stem-like) and evasion of cell death processes. Thus,

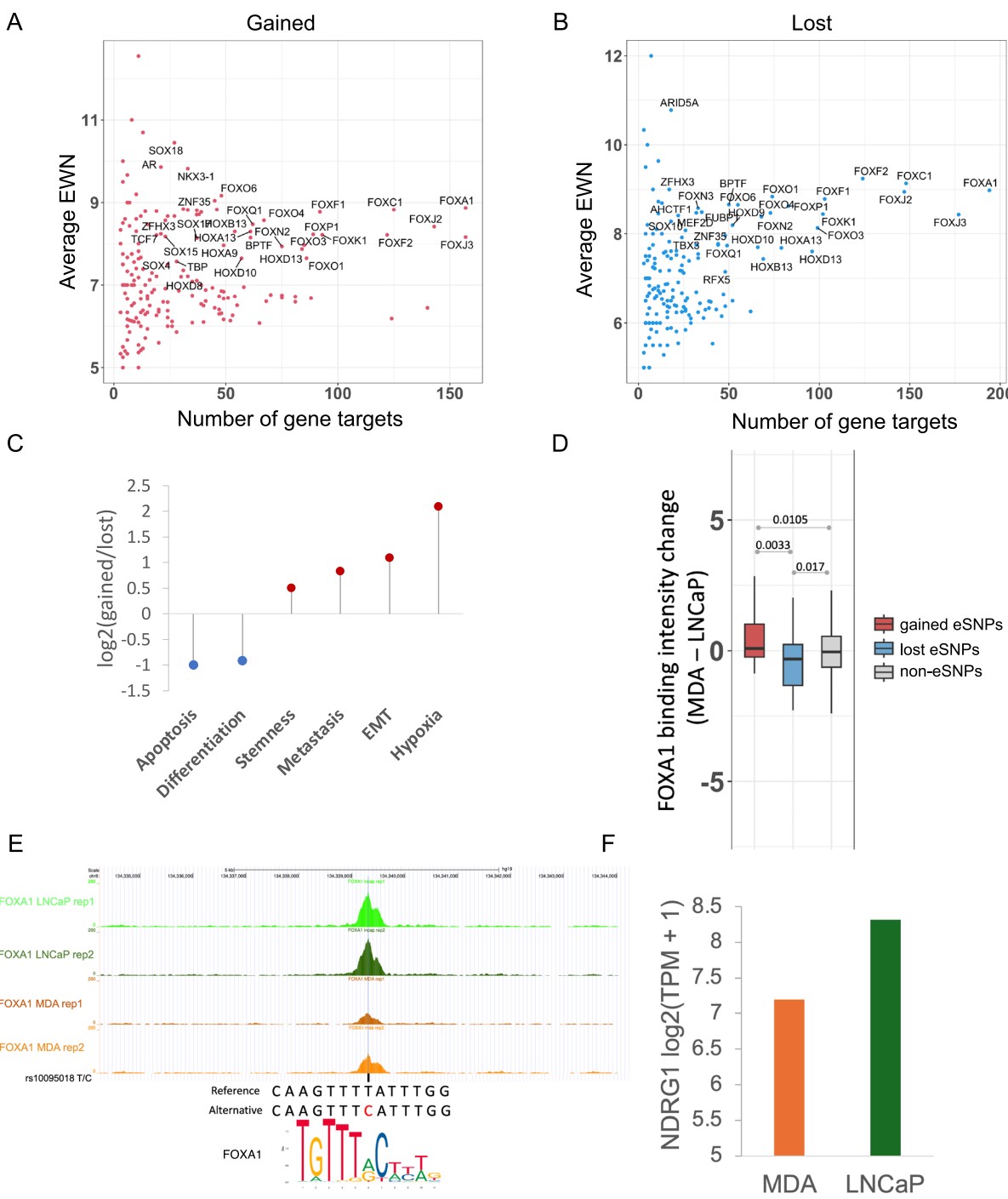

**Fig. 5 | Gained and lost activity eSNPs disrupt binding of FOX TFs and are associated with complementary mechanisms.** Effect size (*y*-axis) of gained (**A**) and lost (**B**) activity eSNPs associated with enriched TFs against their putative target gene counts (*x*-axis). Only the TFs with a number of gene targets ≥15 and an average EWN≥7 are labeled in the figure. **C** Fold enrichment of different cancer signatures in FOXA1-associated gained activity eSNPs compared to FOXA1-associated lost activity eSNPs. **D** FOXA1 binding signal change (MDA - LNCaP) at gained (*n* = 56) and lost (*n* = 62) activity eSNP positions where MDA PCa 2B (dubbed MDA) genotype has a greater number of alternate alleles. The non-eSNP positions (*n* = 7933) are used as background. Bonferroni-corrected *p*-values are obtained from a one-sided Wilcoxon test. In the boxplots, the horizontal line in the middle is the median value, and the lower and upper edges of the boxes correspond to the 25th and 75th percentiles. Extending vertically upwards/downwards of the boxes are the lines showing 1.5 times the interquartile range (i.e., distance between 25th and 75th percentile). **E** Read coverage of FOXA1 at rs10095018 in LNCaP and MDA cell lines, where the alternate allele C disrupts the binding of FOXA1 according to the consensus motif of FOXA1 (see Fig. S5B for larger motif logo). The graphic of the genomic coordinates with the track of FOXA1 reads coverage was generated using UCSC genome browser[69] (http://genome.ucsc.edu/; UCSC hg19 assembly). **F** Gene expression measured as $\log_2(TPM + 1)$ of NDRG1, a likely target of an enhancer harboring rs10095018, in two prostate cancer cell lines: MDA and LNCaP. Source data for these figures is provided as a Source data file.

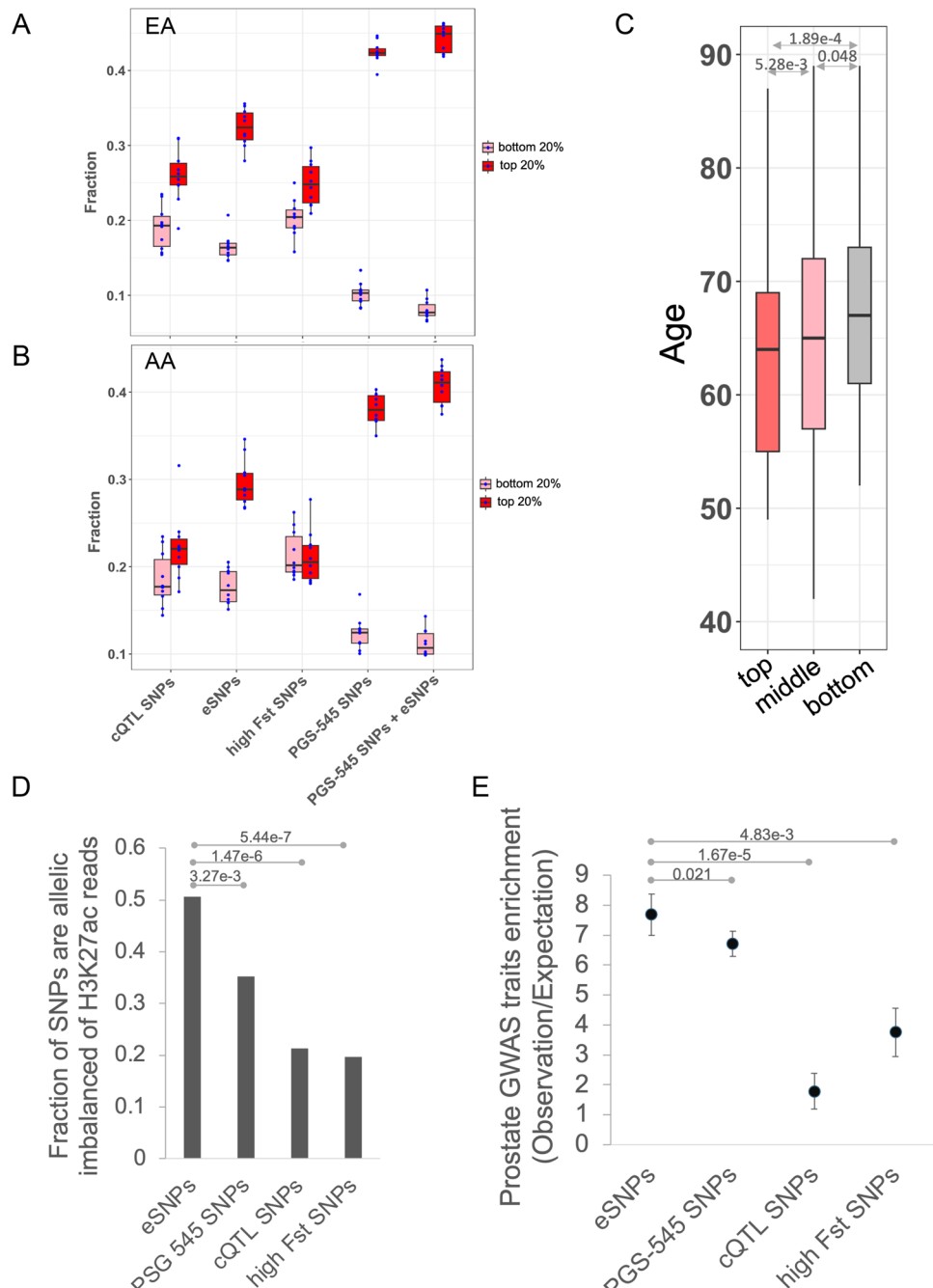

**Fig. 6 | PrCa risk assessment. A** For different SNP sets, the plot shows the percentage of EA cohort test cases that are in the top and the bottom 20 percentile PRS among controls; error bars show the standard deviation across the 10-fold cross-validation ($n = 10$ for each category of each SNP set). **B** For different SNP sets, the plot shows the percentage of AA cohort cases in top and bottom 20 percentile PRS among controls, based on the model trained in EA cohort; error bars show the standard deviation of the 10-fold bootstrapping (90% of the AA cohort), $n = 10$ for each category. **C** Age distribution of the AA PrCa patients with top 10% ($n = 93$), middle 50% ($n = 746$), and bottom 10% ($n = 93$) percentile of eSNP+PGS-545 SNPs-based PRSs. *P*-values are based on a one-sided Wilcoxon test. **D** Fraction of heterozygous SNPs that exhibit allelic imbalance. *P*-values are based on one-sided

Fisher's exact test. **E** Enrichment of prostate-related diseases/traits for different sets of SNPs, which was quantified by the observed ratio of the fraction of SNPs overlapping prostate traits to the fraction of SNPs overlapping other traits, relative to the expected ratio. Central dots are the median values. Error bars show the standard deviation of the 100 bootstrap samples of 90% of the SNPs ($n = 100$ for different sets of SNPs). Bonferroni-corrected *P*-values are obtained based on a one-sided Wilcoxon test. In boxplots (A, B, C), the horizontal line in the middle is the median value, and the lower and upper edges of the boxes correspond to the 25th and 75th percentiles. Extending vertically upwards/downwards of the boxes are the lines showing 1.5 times the interquartile range (i.e., distance between 25th and 75th percentile). Source data for these figures is provided as a Source data file.

the gained and lost activity eSNPs collectively contributing to a PrCa predisposition. The identified eSNPs likely function by disrupting the binding of key TFs, predominantly belonging to the FOX family[36,40]. Although both the gained and the lost activity eSNPs disrupt the binding of the same set of TFs, they regulate different target genes.

The target genes of gained activity eSNPs disrupting FOX binding are involved in oncogenic processes, such as EMT[41], stemness[41], hypoxia[42], and metastasis. Conversely, the target genes of lost activity eSNPs disrupting FOX binding are involved in tumor suppression processes, including differentiation and apoptosis (Fig. 5C). Therefore, loss of

enhancers caused by lost activity eSNPs would lead to inhibition of cell differentiation[43] and evasion of cell death[44]. This aligns with the enrichment of the overall gained and lost eSNPs in the two complementary mechanisms that collectively contribute to PrCa risk (Fig. 4A, B).

While our computational analysis points to the FOX family of TFs, given the well-established role of FOXA1 in prostate development and prostate cancer[36,40], we experimentally validated the predicted effects of the two sets of SNPs on FOXA1 binding in two PrCa cell lines: LNCaP and MDA PCa 2B cells, respectively derived from individuals of European and African ancestry. The MDA PCa 2B cell line, with a higher number of alternate alleles of eSNPs, exhibited increased binding at the gained activity eSNPs and decreased binding at the lost activity eSNPs (Fig. 5D), thereby validating our predictions. The lost activity eSNPs appear to have a larger impact on disrupting FOXA1 binding compared to the gained activity eSNPs (Fig. 5D). This observation is consistent with the fact that TF binding is more easily lost than gained with single nucleotide changes[45].

Unlike traditional GWAS studies that identify risk variants associated with complex human traits or diseases through SNP-trait associations in large cohorts, our approach does not rely on large cohorts with clinical information, in particular, PrCa status, but rather directly identifies potentially functional regulatory SNPs in prostate tissue, which may be relevant to other prostate diseases beyond cancer. Despite not relying on PrCa cohorts, the identified eSNPs add value to the gold standard PGS-545 PrCa variants in discriminating PrCa patients from healthy controls, both within EA and within AA cohort (Fig 6A, B), even though our eSNPs were identified based on differential allele frequency between AA and EA individuals. However, compared to PGS-545, eSNPs are more likely to have a direct functional role beyond mere association with PrCa (Fig. 6D). Interestingly, the genes near PGS-545 SNPs are also associated with developmental processes (Fig. S8), similar to those associated with the lost eSNPs, further substantiating the connection between development and cancer progression. However, the genes near eSNPs are enriched for several additional biological processes likely to be relevant to PrCa, such as telomere maintenance, interferon signaling, and EMT.

Past successes in deep learning models of enhancers[18,46,47] have motivated the current study to focus on non-coding SNPs potentially affecting regulatory enhancer activity. However, in principle, such an approach can be applied to other classes of non-coding SNPs, such as those affecting splicing, as long as accurate sequence-based models capturing those functions can be developed. This provides an avenue for future extension of our study. Likewise, this study can be extended to other cancer types with a substantial hereditary basis, provided the availability of the active enhancer data in the appropriate cell or tissue type. Another point worth noting is that our deep learning model is trained on prostate enhancers, which can affect not just malignant transformation, i.e., PrCa, but other maladies of the prostate, which should be explored in a follow-up study. Much like the QTLs identified by GWAS, the eSNPs identified in our study must be rigorously tested experimentally to establish their causality. As a strength of our study, we have provided initial experimental validation of our predictions, vis-à-vis FOXA1 binding.

## Methods
### Data processing
For H3K27ac and DHS data in the LNCaP cell line, fastq files were downloaded from the European Nucleotide Archive (https://www.ebi.ac.uk/ena/browser/home). Reads were trimmed based on the overlap with the adapter sequence, and those with an average base quality of <20 were removed using Trimmomatic[48] version 0.39. Trimmed reads were aligned to hg19.p13.plusMT.no_alt_analysis_set genome using bowtie2[49] version 2.5.1 (https://bowtie-bio.sourceforge.net/bowtie2/index.shtml) with default parameters. Duplicate reads from the

alignment files were removed, and alignments with a mapping quality score of <20 were discarded using samtools[50] version 1.16.1 (http://www.htslib.org/). Peaks were called using MACS2[51] version 2.2.9.1. For H3K27Ac data, broad peaks were called. For DHS data, narrow peaks were called without building a model, shifting the reads by 150 bp towards the 5′ end and extending by 300 bp towards the 3′ end.

The ChIP-seq experiments of FOXA1 were done in duplicate and we used bamCoverage from deeptools/3.5.1 with a bin size of 25 bp and normalized for read depth using reads per genomic content (RPGC) with an effective genome size of 2.7 billion. We then took the average of the normalized signals of duplicates of each cell line for comparison of signal intensities across the two cell lines.

### Cell lines
LNCaP (cat#CRL-1740) and MDA-PCa-2b (cat#CRL-2422) cell lines were purchased from ATCC and grown in media specified by ATCC: RPMI with 10% fetal calf serum (Gibco) for LNCaP, F-12K Medium with 20% fetal calf serum (Gibco), 25 ng/mL cholera toxin (Sigma), 10 ng/mL mouse EGF (Corning), 0.005 mM phosphoethanolamine (Sigma), 100 pg/mL hydrocortisone (Sigma), 45 nM sodium selenite (Sigma), and 0.005 mg/mL human recombinant insulin (Thermofisher) for MDA-PCA-2b. All cell lines were authenticated by short tandem repeat profiling (Laragen) and tested negative for mycoplasma (ATCC).

### FOXA1 ChIP-seq experiments in two PrCa cell lines
Approximately 10 million cells were fixed using 2 mM DSG (CovaChem cat#1301) for 10 min, followed by 1% formaldehyde for 10 min at 37 °C, and quenched with 2 M glycine for 5min at room temperature. Cross-linked cells were resuspended in cold lysis buffer (50 mM Tris, pH 8.0, 10 mM EDTA, pH 8.0, 0.5% SDS, 1X protease inhibitor cocktail) and sheared using a Bioruptor Pico (Diagenode) device. Fragmented chromatin was incubated with 3 μg of FOXA1 (Abcam, cat#ab23738, lot# 1014292-3) antibody overnight at 4 °C. A fraction of each sample, prior to the addition of antibody, was used as an input control. Protein A/G beads (ThermoFisher cat#10015D) were added and incubated for 1 h at 4 °C, washed six times with RIPA buffer (50 mM HEPES pH 7.6, 1 mM EDTA pH 8.0, 500 mM LiCl, 0.7% sodium deoxycholate, 1% NP40), and eluted in elution buffer (100 mM NaHCO3, 1% SDS). Samples, including input DNA, were treated with RNase A (ThermoFisher cat#EN0531) for 30 min at 37 °C followed by proteinase K (New England Biolabs cat#P8107S) overnight at 65 °C to reverse crosslinking. ChIP DNA was purified with Monarch Genomic DNA Purification Kit (New England Biolabs), and concentrations were quantified by Qubit fluorometer (ThermoFisher) and TapeStation (Agilent). Sequencing libraries were prepared by NEBNext UltraII DNA Library Prep Kit (New England Biolabs cat#E7103S) per manufacturer's instructions and sequenced on the Illumina NextSeq2000 platform at the CCR Genomics Core at the National Cancer Institute, NIH, Bethesda, MD. The quality control measurements of the ChIP-seq experiments are listed in the supplementary information section II.

### A deep convolutional neural network model of prostate enhancers
We built a deep convolutional neural network to predict tissue-specific enhancer activity directly from the enhancer DNA sequence. The deep learning model has the same architecture as the one in our previous study[18], comprising five convolution layers with 320, 320, 240, 240, and 480 kernels, respectively. Higher-level convolution layers receive input from larger genomic ranges and are able to represent more complex patterns than the lower layers. The convolutional layers are followed by a fully connected layer with 180 neurons, integrating the information from the 1 kb sequence. In total, the DLM has 3,631,401 trainable parameters. We used the Python library Keras version 2.1.5 (https://github.com/keras-team/keras) to implement our model. The positive sets contain the 1 kb sequence (and its reverse complement)

of LNCaP enhancers. We used the DHS peaks overlapping H3K27ac peaks to delimit enhancer regions. DHS peaks located within the same H3K27ac peak were merged into one. The enhancers were then these DHS peaks extended to 1 kb from its original center. Enhancers overlapping promoters (including all alternative promoters) and promoters (intervals [−1000 base pairs (bp), +1000 bp] surrounding the transcription start site) were removed from the enhancer set. Overall, we identified 58,266 LNCaP enhancers. The DHS profiles of non-prostate tissues from Roadmap Epigenomics projects[52], which do not overlap the positive sets, were collected as the negative training set of the deep learning model. The reason we used DHS sites not overlapping prostate H3K27ac peaks as negative control regions is that we aim to distinguish prostate-specific enhancers from otherwise accessible and potential enhancers in other tissues. Seventy percent of the data was used for training, 15% is for parameter tuning (on the validation set), and the remaining 15% is used as the testing set. Larger model-predicted score of the genomic sequence corresponds to a higher propensity to be an active tissue-specific enhancer[18]. The genomic sequence with a deep learning model score ≥0.58 (selected based on FPR ≤ 0.01) is predicted to be an active enhancers.

### Effect size of a SNP

We evaluate the effect of an SNP by combining its effects across all possible potential positions of a putative enhancer. For a given SNP, using a stride as 20 bp, there are 21 2-KB windows overlapping the SNP within a (−200 bp, +200 bp) region centered on the SNP. The reason we restricted our sliding windows to cover the central (−200 bp, +200 bp) region is that the alternate alleles tend to have a larger effect on enhancer activity within this narrow region[47] and tend to have more consistent directions of deltas (Fig. 2C). The number of windows in which exactly one of the alternate alleles converts a neutral sequence to an active enhancer and vice versa, according to the model, is termed the essential window number (EWN). The EWN, which ranges from 0 to 21, is utilized to assess the effect size of the SNP on enhancer activity. Further, we expect a consistent directional impact of the SNP on enhancer activity across the 21 windows. Specifically, if the alternate allele exhibits positive delta scores in over 85% of the windows and converts a neutral sequence to an active enhancer in at least 5 windows (EWN ≥ 5) according to our model, the SNP is classified as a gained eSNP, suggesting an enhancer gain. Similarly, if the alternate allele has negative delta scores in over 85% of the windows and changes an active enhancer to a neutral sequence in at least 5 windows (EWN ≥ 5), the SNP is considered a lost eSNP, suggesting an enhancer loss.

### Identification of allelic imbalance in H3K27ac data

We reasoned that if an eSNP locus happens to be heterozygous in a sample, we would expect the two alleles to have differential enhancer activity. Specifically, the alternate alleles of the gained eSNPs should have higher enhancer activity than the reference alleles, and the alternate alleles of the lost eSNPs should have weaker enhancer activity than the reference alleles. This should be reflected in the differential representation of the two alleles among the H3K27Ac reads of the locus. The H3K27ac reads were extracted using BaalChIP[53]. Allelic counts over heterozygous sites of the 5 EA PrCa patients H3K27ac data[7] were merged, and variants that had at least six reads were further processed for allele-specific enhancer activity analysis with a binomial test. We use the heterozygous sites of the gained and lost non-eSNPs as the background. The gained non-eSNPs are high $F_{ST}$ SNPs with positive delta scores, and the lost non-eSNPs are high $F_{ST}$ SNPs with negative delta scores. For a heterozygous site of gained eSNPs, if the ratio of read number of the alternate allele to that of the reference allele is over 1.5 and the binomial $P$ value ≤ 0.01, the position is considered to have allelic imbalance. For a heterozygous site of lost eSNPs, if the ratio of read number of the reference allele to that of the alternate allele is over 1.5 and the binomial $P$ value ≤ 0.01, the position is considered to have

allelic imbalance. Notably, the essential mutations are identified according to the sequence-based deep learning model, suggesting that the alternate alleles at these positions are likely to create or deactivate enhancers. The alternate alleles at these eSNPs coincidently (and independently) exhibit significantly more or less H3K27ac reads than do the reference alleles (termed allelic imbalance), substantiating that the alternate alleles of gained eSNPs are associated with increased enhancer activity, and alternate alleles of lost eSNPs are associated with decreased enhancer activity.

### Functional enrichment analysis using GREAT and cancerSEA

To probe the potential functional roles of gained and lost eSNPs, we tested for functional enrichment among genes near the eSNPs loci using the online Genomic Regions Enrichment of Annotations Tool (GREAT[23]) version 3.0.0 using single-nearest-genes association rule. The GO terms were considered as enriched if it had at least 10 gene hits with overall binomial $p$-value threshold set as 0.01. The randomly sampled 500 K genome-wide non-eSNPs was used as the background. To characterize the cancer progression related functions of the gained and lost eSNPs associated with FOX family TFs, we tested the enrichment of the nearby genes of the eSNPs in the 14 functional states across cancer types (including stemness, invasion, metastasis, proliferation, EMT, angiogenesis, apoptosis, cell cycle, differentiation, DNA damage, DNA repair, hypoxia, inflammation and quiescence) compiled in the cancerSEA database[37]. The enrichment of the targets of gained eSNPs in the genes of a particular functional state relative to those of lost eSNPs was evaluated as $\log_2$(fraction_gained/fraction_lost), and vice versa, where fraction_gained is the fraction of the target genes of gained eSNPs overlapping with the genes of a particular functional state.

### Survival analysis

We used clinical data from TCGA to model the overall survival of PrCa patients using the gene expression as a predictor variable and age as a covariate in the Cox regression to relate gene expression levels to survival outcomes. We used the R library "survival" for this analysis. The HR represents the change in hazard (risk of death) associated with a unit increase in gene expression. Only the HR with significant $p$-values ($p <= 0.05$) were considered in this study. When calculating the enrichment of genes with large HR (HR > 1.5) of AA-dominant gained eSNPs compared to those of all LNCaP, due to small numbers, Fisher's exact test did not yield significance. We therefore applied a bootstrap-based test to estimate the nominal $p$-value of the enrichment. Specifically, we randomly sampled 1097 genes from the 9840 genes associated with all LNCaP enhancers 100 times, and for each sample, calculated the fraction of genes with a high (HR > 1.5, Bonferroni-adjusted $p$-value ≤ 0.05). Overall, the observed fraction of high-HR genes among gained eSNPs (0.0273) falls within the top 0.0189 percentile of the distribution of these 100 sample fractions.

### Logistic regression

We applied the "glm" method from the R package to apply logistic regression to predict gene expression change (adult prostate−fetal prostate) against categories of gained eSNPs, lost eSNPs, random SNPs, and gene expression in fetal prostate.

$$\text{glm}(\Delta g \sim \text{SNP category} + g_{\text{fetal}}, \text{family} = '\text{binomial}') \qquad (1)$$

where $\Delta g$ is the gene expression change in adult prostate compared to fetal prostate ($g_{\text{adult}} - g_{\text{fetal}}$), which is binary (1 for positive change, 0 for negative change). We integrated the fetal prostate gene expression profile (GSM614545[54]) to GTEx and applied quantile-normalized adult and fetal embryonic prostate gene expression and to remove differences across experiments and batch effects. Three different GLMs were applied to the three sets of SNPs (gained eSNPs, lost eSNPs,

and random SNPs), separately. SNP category is binary for each SNP set. $g_{fetal}$ is the expression level of the gene in the fetal prostate. Positive coefficient of each variable indicates a positive correlation between the variable and $\Delta g$.

### Identification of potential TFBSs associated with the gained and lost eSNPs

To identify potential TF binding sites (TFBS), we used FIMO[55] to scan the profiles of binding sites for vertebrate TF motifs in JASPAR[56], CIS-BP[57], SwissRegulon[58], HOCOMOCO[59], and UniPROBE[60] databases, along the 100-bp sequences centered at the gained and lost eSNPs. We identified motif-specific thresholds to limit the FDR no more than five false positives in 10 kb of sequence by scanning each motif on random genomic sequences using FIMO[55]. Enrichment of a motif associated with gained/lost eSNPs (foreground) relative to non-eSNPs (background) was ascertained using Fisher's exact test. The occurrence of a particular TFBS in the set of 100 bp genomic sequences centered by the gained/conserved eSNPs was normalized by the total number of eSNPs. Notably, we included the sequences containing both the reference and alternate alleles to avoid allelic bias.

### Polygenic risk score

Before estimating the PRS, we applied Minimac4 algorithm to impute genotype using Michigan Imputation Server[61]. We applied multivariate linear regression using 10-fold cross-validation in the EA cohort (PEGASUS) to test the predictive accuracy in PrCa predisposition of different sets of risk variants. To estimate the PRS in the independent AA cohort (Ghana) and test the model, we retrain the model using the entire EA cohort and applied it to score the AA individuals using bootstrapping, by randomly subsampling 90% the AA individuals 10 times. The PRS is the weighted sum of the alternative allele number of the risk variants in a given genotype, where the weights are the coefficients of the trained multivariate linear regression model.

### Reporting summary

Further information on research design is available in the Nature Portfolio Reporting Summary linked to this article.

## Data availability

The SNP data are available at 1000 Genome Project (phase 3)[62] (https://www.internationalgenome.org/data-portal/data-collection/phase-3). The LNCaP H3K27ac peaks after DHT treatment used in this study can be found by GSM1249448[21] [https://www.ncbi.nlm.nih.gov/geo/query/acc.cgi?acc=GSM1249448]. The DHT treated DHS data in the LNCaP cell line are available at GSM822388[63] [https://www.ncbi.nlm.nih.gov/geo/query/acc.cgi?acc=GSM822388]. The H3K27ac and ATAC-seq data used to train an enhancer model in normal primary prostate tissues can be obtained from GSE130408[64] [https://www.ncbi.nlm.nih.gov/geo/query/acc.cgi?acc=GSE130408]. The mapped bam files for H3K27ac data of 5 EA PrCa patients for allelic imbalance analysis are provided by the authors of study[7] GSE205885. The gene expression profile of fetal prostate can be found at GSM614545[54] [https://www.ncbi.nlm.nih.gov/geo/query/acc.cgi?acc=GSM614545]. The list of cQTL SNPs were obtained from the authors of[7]. The PGS SNPs from study[8] and can be obtained on the PGS Catalog under accession codes PGP000488 [https://www.pgscatalog.org/]. The genotypes of PrCa cases and controls with European ancestry of the GWAS study (National Cancer Institute (NCI) Prostate Cancer Genome-Wide Association Study for Uncommon Susceptibility Loci (PEGASUS)) can be obtained from dbGap phs000882.v1.p1 [https://www.ncbi.nlm.nih.gov/projects/gap/cgi-bin/study.cgi?study_id=phs000882.v1.p1], and the genotypes of PrCa cases and controls with African ancestry from the study (Ghana)[9] are available at dbGap phs000838.v1.p1.c1 [https://www.ncbi.nlm.nih.gov/projects/gap/cgi-bin/study.cgi?study_id=phs000838.v1.p1]. Access to individual level genotype, sequencing data and detailed phenotype data deposited in dbGap requires permission by dbGap. The processed H3K27ac HiChIP data of LNCaP cell line from the study[65] can be downloaded and visualized interactively via the WashU Epigenome Browser link[66] (https://epigenomegateway.wustl.edu/). The Batch-corrected RNA-seq data for LNCaP and MDA PCa 2b cell lines can be found at Dependency Map (DepMap) portal[67] (DepMap, Broad (2023). DepMap 23Q4 Public. Figshare+. Dataset. https://depmap.org/portal). As there is no effects/dependency Dep-Map score for prostate cancer, we simply took the average DepMap scores across cancer types as an estimate of the DepMap score for each gene. The SNP array-based genotype of LNCaP cell line is obtained from GSM888346, and the SNP array-based genotype of MDA PCa 2b cell line is obtained from GSM888377. The FOXA1 ChIP-seq data generated in this study is deposited at GSE276748. The human GRCh37 (hg19) reference genome is used throughout the study. Clinical data for TCGA PrCa can be downloaded from PanCanAtlas (https://gdc.cancer.gov/about-data/publications/pancanatlas). Gene expression data for TCGA PrCa patients are available from the toil-hub of UCSC-Xena browser (https://xenabrowser.net/datapages/?cohort=TCGA%20Pan-Cancer%20(PANCAN)&removeHub=https%3A%2F%2Fxena.treehouse.gi.ucsc.edu%3A443). Patient information on race, ethnicity, and/or ancestry is self-reported. Source data are provided with this paper.

## Code availability

The tool developed in this study for identifying eSNPs and all the codes used in processing and analysis of datasets are available are deposited at GitHub (https://github.com/hannenhalli-lab/ProSNP-DL), licensed under MIT license (https://github.com/hannenhalli-lab/ProSNP-DL/blob/main/LICENSE.txt). The corresponding DOI is as follows: https://doi.org/10.5281/zenodo.17178792[68].

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

## Acknowledgements

This work used the computational resources of the NIH HPC Biowulf cluster. Next-generation sequencing was done at the CCR Genomics Core at the National Cancer Institute, NIH, Bethesda, MD. We would like to thank Dr. Baca, Sylvan C, Dr. Fortunato, Brad J., Dr. Ziwei Zhang, Dr. Xintao Qiu, and Dr. Mathew Freedman for kindly sharing the supplementary materials of their works with us. We would like to thank Dr. Kun Wang, Dr. Vishaka Gopalan, Annan Timon, and Arati Rajeevan for their feedback. This research was funded, in part, by U.S. National Cancer Institute grant 1-ZIA-BC011979-02 (to S.H.) and 1-ZIA-BC011973-04 (to D.T.).

## Author contributions

S.L.: Data curation, study design, software, formal analysis, investigation, visualization, methodology, writing–original draft, writing–review and editing. K.F.: experiment conduction, formal analysis, visualization, writing–review and editing. N.S., A.S.: Data curation, formal analysis, writing–review and editing. P.S.R, D.N.: supervision, project administration, writing–review and editing. D.Y.T.: supervision, experimental design and supervision, project administration, writing–review and editing. S.H.: supervision, conceptualization, study design, supervision, funding acquisition, investigation, visualization, methodology, writing–original draft, project administration, writing–review and editing.

## Funding

## Competing interests

The authors declare no competing interests.
