## [Peer Review File · Nature Communications]

Non-coding genetic variants underlying higher prostate cancer risk in men of African ancestry

Corresponding Author: Dr Sridhar Hannenhalli

Version 0:

Reviewer comments:

Reviewer #1

(Remarks to the Author)

What are the noteworthy results?

African American (AA) men carry a disproportionate share of prostate cancer (PrCa) morbidity and mortality. Despite this, significant genomic determinants of PrCa have been found primarily in men of non-African ancestry, especially in men of European descent (EA). The authors identified non-coding regulatory polymorphisms that associate with PrCa in AA men with the assistance of a sequenced-based deep learning model of prostate cancer. Furthermore, the authors identified enhancer SNPs (eSNPs) that include the suppression of immunity and inhibition of apoptosis.

Will the work be of significance to the field and related fields?

Yes, I believe this work is of significance to the genomic epidemiology of PrCa in AA who are significantly understudied and under-reported in the literature. A better understanding of these eSNPs could lead to better precision medicine options for AA who carry a disproportionate burden of PrCa.

How does it compare to the established literature? If the work is not original, please provide relevant references.

Established literature has many genomic studies including GWAS and sequenced-based studies of PrCa. This manuscript is novel because of the approach it uses to identify putative genomic determinants, specifically eSNPs of PrCa in AA. This manuscript attempts to contribute towards elucidating why AA men suffer from more aggressive PrCa and are more likely to die from this disease.

Does the work support the conclusions and claims, or is additional evidence needed?

I do believe the work described in this manuscript supports the conclusions and claims met.

Are there any flaws in the data analysis, interpretation and conclusions? Do these prohibit publication or require revision?

I do find a few items that I believe should be addressed by the authors:

1) The authors identified regulatory variants associated with divergent PrCa risk in AA and EA Minor Allele Frequency; MAF $\geq 5\%$ in either the EA or the AA population. I think the authors should explain why the MAF $\geq 5\%$ was for either EA or AA. Since this study's objective was to focus on AA men with PrCa, I would think that they would begin with focusing on MAF that were $\geq 5\%$ in AA only and then filter these SNPs further keeping the most disparate allele frequencies between AA and EA.

2) The definition of essential window number (EWN) is abbreviated early on in the manuscript before it is defined later in the manuscript. The explanation of EWN should be provided prior to Figure 1. I believe the readership of this journal who are unfamiliar with genetic epidemiology methods would benefit from a more in-depth explanation of EWN and why it was useful

for identifying eSNPs.

3) To me, Figure 3C is confusing. Figure 3C appears to show EA dominant and not AA dominant. The authors should re-evaluate this figure.

Is the methodology sound? Does the work meet the expected standards in your field?

Yes, I believe the methodology is sound and well-described. I like the detailed figures that help to convey the methodology.

Is there enough detail provided in the methods for the work to be reproduced?

Yes, I do believe that the authors provide provided a significant amount of detail for their work to be reproduced.

(Remarks on code availability)

Reviewer #2

(Remarks to the Author)

The manuscript by Shan Li et al addresses enhancer variants (eSNPs) presenting extreme heterogeneity in frequency between African and European men, and evaluates their underlying risk for prostate cancer, which is known to have higher incidence in the former ancestry. This is a very complete manuscript, combining mining multi-omics databases in cancer cell lines/patient/GWAS cohorts' data with confirming results by experimental validation (new results have been made publicly available). Also, they explore deeply the data by applying diverse and informative methods, whose pipeline is made available at GitHub. We agree with publication, after the authors address the following comments:

1- Authors should state in a very clear way which EA and AA populations they used to infer the "10,171,946 non-coding SNPs that were common (Minor Allele Frequency; MAF \geq 5%)". And also include how many individuals were considered in those cohorts. If the 1000 Genomes European and Sub-Saharan African cohorts were not used to provide this information, please include explanation for not considering it, when this database is widely used as the reference for worldwide population groups.

2- Although there are not many prostate cancer cell lines, there are a few Europeans and one African. Given this fact, it is difficult to understand why the authors limited their deep learning model to be applied to the European LNCaP cell line. Basing their entire results in only one cancer cell line can be potentially biasing when extrapolating for overall prostate cancer. Authors should also run the model in the African MDA PCa 2B cancer cell line, and compare overlapping between both. If data are not available for the African cancer cell line, consider any other European ones would be more robust than doing so only in one.

3- Related with the fact that the authors identified eSNPs playing a role in overall prostate function and not only in prostate cancer, did authors consider to use H3K27ac ChIP-seq data available for normal prostate tissue? This information could help in identifying the active enhancers that are prostate cancer related, and clean out other non-cancer related ones. This added information could render more informative the PGS for eSNPs. Please check availability of data on normal prostate tissue in Pomerantz et al. 2023 (<https://pubmed.ncbi.nlm.nih.gov/32690948/>) and consider to do this extra analysis for PGS calculations.

4- In fact the H3K27ac ChIP-seq data for the normal prostate tissue could also be used to run the deep learning model, and compare it with data obtained for the cancer cell lines. Thus, from the beginning active enhancers of prostate tissue could be distinguished from active enhancers for prostate cancer.

5- Minor comments: Please spell out raQTL and cQTL. DepMap on page 19 is written many times as DapMap.

(Remarks on code availability)

Reviewer #3

(Remarks to the Author)

(Remarks on code availability)

Reviewer #4

(Remarks to the Author)

Summary

Li et al. present a set of ~2k non coding SNPs particularly present in African ancestry(AA) people with respect to European ancestry (EA) people, which is biologically significant in the context of Prostate cancer. While AA is known to be associated with higher PrCA incidence and lower survival rates, the role of non coding SPNs have not been thoroughly investigated in this specific case.

The authors identify the said SNPs using a Deep Learning model trained to predict the potential change in an enhancer activity due to a single nucleotide change in an enhancer sequence. They further group the found SNPs into “gained” and “lost”, based on the increased/decreased enhancer activity by the SNP. The author first validated this classification using acetylation of histone H3K27. They then compare AA-dominant and EA-dominant SNPs based on their correlation with PrCa risk.

The authors discuss the role of the found SNPs based on their functional characterization and they propose that “gained” and “lost” SNPs work in a complementary way, increasing the activity of oncogenic pathways while decreasing pathways involved in tumor suppression.

Finally, they show that by building a risk core built upon the found SNPs, they are able to improve previously reported gold standard methods.

= Major comments

- Cell Line Selection and Potential Bias

The authors evaluate the impact of eSNPs on FOX binding using two cell lines from different cancer types. Could this difference introduce bias in the results? How much these cell lines represent the extent of germline variability of the target populations? Could other public databases be included in this analysis? The authors should discuss the potential implications of this experimental choice and whether it may influence the conclusions drawn.

- Statistical Tests for p-values

Numerous figures present p-values; however, the statistical tests employed in their calculation are frequently not disclosed. This omission hinders the evaluation of the proposed methodologies' robustness and undermines the findings' reproducibility and transparency. For instance, in Figures 1A-B, the specific tests utilised are not indicated, while they are in other figures 6C

Some figures present multiple p-values from repeated statistical tests (e.g., Fig. 4A-B). However, it is unclear how the authors account for the multiple comparison problem. The authors should describe whether any correction method (e.g., Bonferroni, FDR) was applied and, if not, justify why such corrections were unnecessary.

- Reproducibility

It is not clear if the authors provide access to the code or not. To be precise, the proposed architecture has been already published (ref 18). The GitHub provides access to training and test for the DL model. However, there is no script to replicate the analysis presented in the paper, which seems odd. In the attached files the authors say that they will make the SW open source, but it's not clear what SW they refer to.

= Minor comments

- Definition of “DL score”: The term "DL score" appears multiple times in the paper, likely referring to "Deep Learning score", but it is never explicitly defined. The authors should provide a clear definition of this term upon first mention to avoid ambiguity.

- Clarification of Figures 3C, 3D, and S1C: The method used to generate Figures 3C, 3D, and S1C is unclear. The authors should elaborate on the methodology in the Methods section to ensure reproducibility. Additionally, would it be possible to apply a statistical test to these figures and report a p-value to support the findings quantitative

= Overall comment

This paper presents a method for understanding the role of non-coding SNPs in prostate cancer (PrCa), with potential applications to other tumor types. However, it is essential to critically evaluate this research, as fundamental aspects need clarification. While the study has strengths—such as advancing biological understanding and providing potential for improved risk metrics—it is crucial to address the underlying mechanisms and limitations that may affect the interpretation of the findings. Greater clarity on these issues is necessary to ensure that the advancements in risk assessment truly lead to better identification of high-risk patients, thereby reinforcing this work's scientific and clinical value.

(Remarks on code availability)

Reviewer #5

(Remarks to the Author)

(Remarks on code availability)

Reviewer #6

(Remarks to the Author)

Non-coding genetic variants underlying higher prostate cancer risk in men of African ancestry

In this paper the authors employed a sequence-based deep learning model of prostate regulatory enhancers and identified ~2,000 SNPs with higher alternate allele frequency in men of African ancestry that potentially affect enhancer function associated with prostate cancer susceptibility. They explored potential mechanisms associated with the identified enhancer SNPs (eSNPs) that may influence PrCa development and found that the eSNPs could potentially disrupt binding of known prostate transcription factors. They also identified eSNPs that can be combined into a polygenic risk score to add value to current GWAS-based risk variants in assessing PrCa risk in independent cohorts.

Whilst the study is potentially of interest it appears to lack comprehensive training sets to inform 'a deep learning model of enhancer activity' and lacks robust validation using H3K27ac ChIP-seq data from different prostate cell lines. In addition, there are no functional validation experiments to support the proposed effects of eSNPs on their gene targets.

Major criticisms:

1. In developing the machine learning algorithm, the authors only use the one cell line (LNCaP) H3K27ac ChIP-seq data for both training and testing for prostate enhancer prediction. This cell line is not from a normal prostate cell but from an androgen-dependent human prostatic carcinoma cell line originally isolated from a lymph node metastatic lesion and does not represent the diversity of enhancer usage in prostate cells. Expanding the training set to include multiple prostate cell lines would improve the robustness of the model to predict disruption of enhancer function.
2. Similarly, validating that the tool in other prostate cell lines would be more robust than validating in the same cell line. This is feasible given the wide availability of H3K27ac data sets across different prostate cancer cell lines including both androgen-dependent and independent cells.
3. Critically, the study is lacking functional validation experiments to confirm the effects of identified eSNPs on their proposed gene targets. This could be addressed for example using CRISPR at candidate eSNP loci in LNCaP cells or a 3C investigation at candidate eSNP loci to explore disruption of enhancer promoter loops.

Other important analyses:

4. Please provide statistics for Fig 3E. Fraction of genes >1.5 HR for gained eSNPs is very close to the value for all LNCaP enhancers which diminishes any specific effects of gained eSNPs on HR. Also, the related figure Supp 1D shows a similar fraction of genes with >1.5 HR for EA lost and gained SNPs. This seems contradictory to the finding in Figure 3D which shows EA lost and gained eSNPs are not associated with prostate cancer GWAS traits.
5. For the GO analysis the methods state on line 558 - 'To probe the potential functional roles of gained and lost eSNPs, we tested for functional enrichment among genes near the eSNPs loci using the online Genomic Regions Enrichment of Annotations Tool (GREAT23) version 3.0.0 using two-nearest-genes association rule... The whole genome region was used as the background.' -Whole genome is not the correct background. The authors need to control for genomic biases in location of eSNPs or SNPs, eg using genes next to EA dominant eSNPs or non eSNPs as background.
6. In Figure 4D, why do genes proximal to gained eSNPs exhibit significantly higher expression in fetal prostate if they are regulating immune function? All of the pathways highlighted for gained eSNPs are all essential in adult function too.
7. Line 232 - 'Taken together, these results strongly suggest that by activating oncogenic processes and disrupting tumour suppression mechanisms, the gained and lost activity eSNPs collaboratively promote carcinogenesis - This is not supported by the GO analysis for gained eSNPs which mainly includes immune and lipid metabolism terms. Similarly for lost eSNPs, there is a large conceptual gap between lost SNPs being near genes that regulate differentiation development and lost SNPs being involved in disrupting 'tumour suppression mechanisms'. Also see point about line 263 below. Please modify the language of the conclusions over to reflect the actual data.
8. Line 248 - 'Likely to be mediated' is also too strong as only the author refer to only one reference (ref 17) that has shown that one SNP acts via this mechanism but did not provide functional evidence themselves.
9. Line 263 - To the best of my understanding the CancerSEA signatures all pertain to functional cancer states. The

CancerSEA publication mentions nothing about some states being 'pro-tumor' vs 'anti-tumour', rather, the authors have just ascribed these descriptions to some of the states without explaining how or why. The same occurs in lines 389-392 in the discussion.

10. Figure 5D and 5E – Authors should provide details on how they performed FOXA1 ChIP-seq and analyses; it is not mentioned in the results or methods and therefore it is not possible to determine if the claims of loss/gain of peaks can be supported by the data presented. Were the ChIPs performed in triplicates, how is the ChIP-seq data quantitative, ie were spike ins added or normalization performed to allow comparison of signal intensities across samples? Did the ChIPs work effectively in both cell-lines to permit a quantitative comparison of FOXA1 binding efficiencies? Further to this, the candidate screenshot in 5E shows a very zoomed in area of the peak. It is important to provide QC data to determine that the FOXA1 signal genome-wide is limited to FOXA1 binding sites and is absent elsewhere in the genome as a testament to the quality of the data.

(Remarks on code availability)

I don't have the expertise to review the code.

Version 1:

Reviewer comments:

Reviewer #1

(Remarks to the Author)

I believe the results of this revised manuscript remain noteworthy. The authors identified non-coding regulatory polymorphisms associated with PrCa in African American (AA) men. Men of AA ancestry shared a disproportionate burden of PrCa morbidity and mortality and are underrepresented in PrCa genomics research. They identified enhancer SNPs (eSNPs) which are associated with apoptosis and immunosuppression which have significant roles in cancer development and progression.

I concur with the authors that the role of non-coding regulatory polymorphisms in PrCa has not fully investigated. I also believe this report could add a novel aspect of including eSNPs in the construction of ancestry-dependent polygenic risk scores for PrCa.

This revised work is more comprehensive. In my view, the authors have taken great care to sufficiently address the concerns of all reviewers. The manuscript would add to the existing literature in the area of PrCa genomics. The methodology is sound and meets the expected standards in the PrCa genomics field. I believe this revised manuscript also provides enough detail in the methods to be reproducible.

Overall, I do not detect any unresolved and/or newly recognized major flaws in the revised manuscript's methodology, data analysis, interpretations and conclusions. I do not believe that this manuscript requires further revision, and I think it is now suitable for publication.

(Remarks on code availability)

N/A

Reviewer #2

(Remarks to the Author)

I am satisfied with the reply of the authors to my comments. Their confirmation that the results closely resemble those of other prostate cancer cell lines and normal primary prostate tissues, and the inclusion of this information in the manuscript, render the investigation more robust.

(Remarks on code availability)

Reviewer #3

(Remarks to the Author)

(Remarks on code availability)

Reviewer #4

(Remarks to the Author)

The reviewers have answered our concerns and improved the current version of the manuscript.

(Remarks on code availability)

Reviewer #5

(Remarks to the Author)

(Remarks on code availability)

The code of the model is present but not the full code to reproduce the analysis.

Reviewer #7

(Remarks to the Author)

In this manuscript Li et al. have aimed to interrogate single-nucleotide polymorphisms (SNPs) underlying increased prostate cancer risk and mortality in men of African-American ancestry. The authors approached this question by employing a sequence-based deep learning model of regulatory H3K27ac enhancers associated with ~2000 eSNPs that are present at a higher alternate frequency in AA men and potentially associated with PC susceptibility. The authors noted that eSNPs tend to disrupt the binding of FOX, AR and HOX families.

Overall, the manuscript is interesting, comprehensive and is systematic in investigating the role of nucleotide polymorphisms. The authors have also addressed a majority of the concerns of the previous reviewers.

Comments:

Line 26: The authors mention the eSNP disrupt the binding of known prostate TFs including FOXA1, AR and HOXB13. Also, Line 274 of the manuscript mentions the same. In Line 277, the disruption is described as (gain or loss) of TF binding. The term "disrupt" is confusing, as it would align more with loss of function of TF binding and tends to deemphasize the gain of TF activity at H3K27ac enriched eSNPs. A revision of this term is suggested.

From the mechanistic/validation perspective the manuscript has some weaknesses as also pointed out by the earlier reviewer. For e.g in Fig 5E, the eSNP at rs10095018 reduces but not completely disrupts FOXA1 binding in MDA-PCa2B cell line (AA ancestry) compared to LNCaP cell line (EA ancestry). There is no experimental data provided for AR or HOXB13 TF binding at these H3K27ac eSNPs. Likewise, whether AR and HOXB13 binding is affected due to disrupted FOXA1 at eSNP is not demonstrated. Lacking is the important ChIP-seq data for H3K27ac/FOXA1/AR/HOXB13 at this site in these PC two cell lines under the same experimental conditions, or in primary prostate cancer tissues of EA and AA origins, to demonstrate that this indeed there is overlap of eSNP associated with the enhancer activity and TF binding. Intriguingly, the percentage of potential FOXA1 targets is comparable in LNCaP (59.68% and 59.77%) and MDA-PCa2b (59.99% and 64.21%). Therefore, it is not clear whether FOXA1 binding alone can explain the dominant activity of eSNPs in AA versus EA men. This aspect requires revision or further clarification by authors to support their claims.

(Remarks on code availability)

Not tested.

Reviewer's Comments:

We are grateful to the editor and the reviewers for their input and suggestions, which were invaluable in further strengthening our study. Please see below for a point-by-point response to the reviewers' comments.

Reviewer #1 (Remarks to the Author)

What are the noteworthy results?

African American (AA) men carry a disproportionate share of prostate cancer (PrCa) morbidity and mortality. Despite this, significant genomic determinants of PrCa have been found primarily in men of non-African ancestry, especially in men of European descent (EA). The authors identified non-coding regulatory polymorphisms that associate with PrCa in AA men with the assistance of a sequenced-based deep learning model of prostate cancer. Furthermore, the authors identified enhancer SNPs (eSNPs) that include the suppression of immunity and inhibition of apoptosis.

Will the work be of significance to the field and related fields?

Yes, I believe this work is of significance to the genomic epidemiology of PrCa in AA who are significantly understudied and under-reported in the literature. A better understanding of these eSNPs could lead to better precision medicine options for AA who carry a disproportionate burden of PrCa.

How does it compare to the established literature? If the work is not original, please provide relevant references.

Established literature has many genomic studies including GWAS and sequenced-based studies of PrCa. This manuscript is novel because of the approach it uses to identify putative genomic determinants, specifically eSNPs of PrCa in AA. This manuscript attempts to contribute towards elucidating why AA men suffer from more aggressive PrCa and are more likely to die from this disease.

Does the work support the conclusions and claims, or is additional evidence needed?

I do believe the work described in this manuscript supports the conclusions and claims met.

Are there any flaws in the data analysis, interpretation and conclusions? Do these prohibit publication or require revision?

I do find a few items that I believe should be addressed by the authors:

We sincerely appreciate your kind remarks and are encouraged to see that our findings resonated with you. Please see below for a point-by-point response.

1) The authors identified regulatory variants associated with divergent PrCa risk in AA and EA (Minor Allele Frequency; MAF \geq 5%) in either the EA or the AA population. I think the authors should explain why the MAF \geq 5% was for either EA or AA. Since this study's objective was to focus on AA men with PrCa, I would think that they would begin with focusing on MAF that were \geq 5% in AA only and then filter these SNPs further keeping the most disparate allele frequencies between AA and EA.

This is a very good point and something we had considered. However, because a priori we were equally interested in EA-dominant SNPs where minor allele frequency is lower in AA (and hence may go undetected), as those EA-dominant (or equivalently, AA-depleted) could also reveal major functional differences between EA and AA associated with PrCa. However, our analyses revealed that AA-dominant SNPs indeed have a larger impact on PrCa onset. This observation would have been lost if we had excluded EA-dominant SNPs a priori. We have now made this justification more explicit (Page 8, Line 170-172): “A priori, we are equally interested in both AA-dominant and EA-dominant e-SNPs, as they could both reveal major functional differences between EA and AA associated with PrCa.”.

2) The definition of essential window number (EWN) is abbreviated early on in the manuscript before it is defined later in the manuscript. The explanation of EWN should be provided prior to Figure 1. I believe the readership of this journal who are unfamiliar with genetic epidemiology methods would benefit from a more in-depth explanation of EWN and why it was useful for identifying eSNPs.

We regret this oversight. We have added the rationale of essential window number and rephrased the legend of Figure 1 at line 107-111 on Page 5 “The deep learning model integrated with a sliding window strategy was then applied to the retained 491K SNPs to select the ~2000 AA-dominant SNPs (eSNPs) at which the model predicts substantially different enhancer activities for the two alleles, using a criterion called essential window number (EWN) that measures consistency of the directional impact of functional SNPs on enhancer activity (Methods)”.

3) To me, Figure 3C is confusing. Figure 3C appears to show EA dominant and not AA dominant. The authors should re-evaluate this figure.

Sorry for the misunderstanding; the labels are above the figure. In Figure 3C and D, we demonstrated that the AA-dominant eSNPs are more likely to have stronger PrCa GWAS signal (the z-score normalized GWAS signal was provided by the original study (Schumacher, F.R. *et al.*)), as compared to the AA-dominant non-eSNPs (Figure 3C), at varying cutoff of the GWAS signal (x-axis). Whereas this is not the case for EA-dominant (Figure 3D). The enrichment (y-axis) was measured as the ratio of the fraction of eSNPs with PrCa GWAS signal z-score greater than the signal cutoff (x-axis) to the fraction of non-eSNPs with PrCa GWAS signal z score greater than the same cutoff (x-axis). So overall, one can observe an increased enrichment pattern of both gained and lost AA-dominant eSNPs in PrCa GWAS signal across increasing cutoff (Figure 3C), but this is not true for EA-dominant eSNPs (Figure 3D). We have reworded this part in the legend of Figure 3 and added a clarification statement on Page 8 (line 183-187) to make it more explicit.

Is the methodology sound? Does the work meet the expected standards in your field?

Yes, I believe the methodology is sound and well-described. I like the detailed figures that help to convey the methodology.

Is there enough detail provided in the methods for the work to be reproduced?

Yes, I do believe that the authors provide provided a significant amount of detail for their work to be reproduced.(Remarks on code availability)

Reviewer #2 (Remarks to the Author)

The manuscript by Shan Li et al addresses enhancer variants (eSNPs) presenting extreme heterogeneity in frequency between African and European men, and evaluates their underlying risk for prostate cancer, which is known to have higher incidence in the former ancestry. This is a very complete manuscript, combining mining multi-omics databases in cancer cell lines/patient/GWAS cohorts' data with confirming results by experimental validation (new results have been made publicly available). Also, they explore deeply the data by applying diverse and informative methods, whose pipeline is made available at GitHub. We agree with publication, after the authors address the following comments:

We sincerely appreciate your encouraging comments.

1- Authors should state in a very clear way which EA and AA populations they used to infer the “10,171,946 non-coding SNPs that were common (Minor Allele Frequency; MAF \geq 5%)”. And also include how many individuals were considered in those cohorts. If the 1000 Genomes European and Sub-Saharan African cohorts were not used to provide this information, please include explanation for not considering it, when this database is widely used as the reference for worldwide population groups.

Yes, we indeed used the SNP data from 1000 Genome Project (phase 3). The 1000 Genomes European and Sub-Saharan African cohorts were used. This is stated in the method section at Line 481 (Page 21).

2- Although there are not many prostate cancer cell lines, there are a few Europeans and one African. Given this fact, it is difficult to understand why the authors limited their deep learning model to be applied to the European LNCaP cell line. Basing their entire results in only one cancer cell line can be potentially biasing when extrapolating for overall prostate cancer. Authors should also run the model in the African MDA PCa 2B cancer cell line, and compare overlapping between both. If data are not available for the African cancer cell line, consider any other European ones would be more robust than doing so only in one.

See response to comment-4.

3- Related with the fact that the authors identified eSNPs playing a role in overall prostate function and not only in prostate cancer, did authors consider to use H3K27ac ChIP-seq data available for normal prostate tissue? This information could help in identifying the active enhancers that are prostate cancer related, and clean out other non-cancer related ones. This added information could render more informative the PGS for eSNPs. Please check availability of data on normal prostate tissue in Pomerantz et al. 2023 (<https://pubmed.ncbi.nlm.nih.gov/32690948/>) and consider to do this extra analysis for PGS calculations.

See response to comment-4.

4- In fact the H3K27ac ChIP-seq data for the normal prostate tissue could also be used to run the deep learning model, and compare it with data obtained for the cancer cell

lines. Thus, from the beginning active enhancers of prostate tissue could be distinguished from active enhancers for prostate cancer.

The following response combines the responses to the related comments from 2-4 of Reviewer #2 and responses to the related comments from 1-2 of Reviewer #6:

Thanks for these suggestions! We applied the LNCaP enhancer model to prioritize non-coding genetic variants under the assumption that the enhancer sequence features of LNCaP cells closely resemble those of other prostate cancer cell lines and normal primary prostate tissues. Also, the 6th reviewer suggested that we expand the training set to include multiple prostate cell lines to improve the robustness of the model. Toward this, we evaluated the robustness of the LNCaP model across different PrCa cell lines and normal prostate tissues.

Training our model requires context-specific H3K27ac data. As for other PrCa cell lines, we have not been able to find the H3K27ac data of the MDA PCa 2B cell line, but we did find the data in VCaP. Our new analyses suggest that the sequence features of VCaP align well with those of LNCaP to a large extent (Response Figure 1AB); specifically, the accuracy (auROC) of predicting LNCaP enhancers using VCaP enhancer model is 0.88 (Response Figure 1A), while the auROC for predicting VCaP enhancers using LNCaP enhancer model is 0.87 (Response Figure 1B). To further examine whether eSNPs exhibit consistent effect sizes in modulating enhancer activity (measured as average delta across all sliding windows) using different models, we applied the VCaP model to score the 2,000 eSNPs and compared the delta values with those obtained using the VCaP-based model. The two resulting average delta scores exhibited a strong correlation with those generated by the LNCaP model, with a Spearman correlation of 0.754 (Response Figure 1C).

We also evaluated the accuracy of the primary prostate enhancer model in predicting LNCaP enhancers and vice versa. Additionally, we construct a prostate integrative enhancer model by merging all open chromatin regions (DHS or ATAC-seq) and H3K27ac data from the two prostate cancer cell lines (LNCaP and VCaP) and normal primary prostate tissues. This model was built using 1KB merged open chromatin regions (DHS or ATAC-seq) that overlap with merged H3K27ac peaks. The goal was to test whether the LNCaP enhancer model could obtain high accuracy across multiple prostate cell lines including the integrative enhancer set. First, using the H3K27ac and ATAC-seq data of Pomerantz et al. 2023, as suggested, to train an enhancer model in normal primary prostate tissues, we observed that the auROC for predicting LNCaP enhancers was 0.72 (Response Figure 1A). Moreover, the auROC for predicting primary prostate enhancers using the LNCaP model was 0.8. More importantly, the LNCaP enhancer model could accurately predict the prostate integrative enhancer regions (auROC = 0.87), indicating that the enhancer model based on LNCaP cell line is able to effectively capture prostate-specific sequence features.

In addition, the delta scores of eSNPs derived from LNCaP enhancer model correlates well (Spearman correlations = 0.75, 0.56, 0.68) with those based on other prostate enhancer models (Response Figure 1C-E).

Therefore, in summary, using the LNCaP model to prioritize SNPs is robust and generally applicable for our purposes. We have put this part in the supplemental information section I (Page 28-29) and mentioned it in the main text at line 126-129, page 5.

Response figure 1. **Robustness of enhancer model.** A) auROC of models trained on VCaP, normal primary prostate enhancer, or prostate integrated enhancers (enhancers merged from LNCaP, VCaP, and primary prostate) models on LNCaP enhancers. B) auROC of LNCaP-trained enhancer models on VCaP, primary prostate enhancers, and prostate integrated enhancers. C) scatter plot of delta scores of eSNPs using VCaP-trained enhancer model against those using LNCaP-trained enhancer model. D) scatter plot of delta scores of eSNPs using normal prostate enhancer-trained model against those based on LNCaP-trained enhancer model. E) scatter plot of delta scores of eSNPs using prostate integrated enhancers-trained model against those using LNCaP-trained enhancer model.

5- Minor comments: Please spell out raQTL and cQTL. DepMap on page 19 is written many times as DapMap.(Remarks on code availability)

We regret the oversight. We have now included the full name for raQTL (Line 124, page 5), cQTL (Line 362, page 16), and DepMap (Line 495, page 20).

Reviewer #3 (Remarks to the Author)

I co-reviewed this manuscript with one of the reviewers who provided the listed reports. This is part of the Nature Communications initiative to facilitate training in peer review and to provide appropriate recognition for Early Career Researchers who co-review manuscripts.(Remarks on code availability)

Reviewer #4 (Remarks to the Author)

Summary

Li et al. present a set of ~2k non coding SNPs particularly present in African ancestry(AA) people with respect to European ancestry (EA) people, which is biologically significant in the context of Prostate cancer. While AA is known to be associated with higher PrCA incidence and lower survival rates, the role of non coding SPNs have not been thoroughly investigated in this specific case.

The authors identify the said SNPs using a Deep Learning model trained to predict the potential change in an enhancer activity due to a single nucleotide change in an enhancer sequence. They further group the found SNPs into “gained” and “lost”, based on the increased/decreased enhancer activity by the SNP. The author first validated this classification using acetylation of histone H3K27. They then compare AA-dominant and EA-dominant SNPs based on their correlation with PrCa risk.

The authors discuss the role of the found SNPs based on their functional characterization and they propose that “gained” and “lost” SNPs work in a complementary way, increasing the activity of oncogenic pathways while decreasing pathways involved in tumor suppression.

Finally, they show that by building a risk core built upon the found SNPs, they are able to improve previously reported gold standard methods.

Major comments

- Cell Line Selection and Potential Bias

The authors evaluate the impact of eSNPs on FOX binding using two cell lines from different cancer types. Could this difference introduce bias in the results? How much these cell lines represent the extent of germline variability of the target populations? Could other public databases be included in this analysis? The authors should discuss the potential implications of this experimental choice and whether it may influence the conclusions drawn.

We regret lack of clarity. To be sure, the two cell lines are not from two different cancer types, instead, they are both derived from human prostate cancer cell lines but from individuals of two different ancestries. Specifically, the LNCaP cell line is derived from a prostate adenocarcinoma from a 50-year-old Caucasian male (Horozewicz, J.S. *et al*), and the MDA PCa 2B is derived from a prostate adenocarcinoma from a 63-year-old African American male (Navone, N.M. *et al.*). The fact that MDA PCa 2B has an overall greater number of alternate alleles at eSNP sites than LNCaP aligns perfectly with the expectations that these eSNPs are African American dominant, which further indicates these two cell lines indeed represent the germline variability of the corresponding populations. In addition, the impact of eSNPs on differential FOXA binding between the two cell lines have nothing to do with underlying populations but simply the genotypic difference at these eSNPs sites between the two cell lines. We have edited this section for improved clarity (Line 320-328, Page 15): “*To experimentally validate the impact of eSNPs on FOX binding, we performed ChIP-seq for FOXA1 in two PrCa cell lines of distinct ancestral origins: LNCaP cells, derived from a lymph node metastasis of a 50-year-old Caucasian male with prostate adenocarcinoma¹⁹, and MDA PCa 2B cells, isolated from the prostate of a 63-year-old Black male patient with adenocarcinoma²⁰ (Methods). First, consistent with our expectation, MDA PCa 2B has an overall greater number of alternate alleles at eSNP sites, aligning with the fact that these eSNPs are AA dominant (Table S5). Furthermore, we observed that relative to LNCaP, FOXA1 binding intensity was greater in MDA PCa 2B at gained activity eSNP sites and lower at lost activity eSNP sites (Fig. 5D, Methods).*”.

- Statistical Tests for p-values

Numerous figures present p-values; however, the statistical tests employed in their calculation are frequently not disclosed. This omission hinders the evaluation of the proposed methodologies' robustness and undermines the findings' reproducibility and transparency. For instance, in Figures 3A-B, the specific tests utilised are not indicated, while they are in other figures 6C

Some figures present multiple p-values from repeated statistical tests (e.g., Fig. 4A-B). However, it is unclear how the authors account for the multiple comparison problem. The authors should describe whether any correction method (e.g., Bonferroni, FDR) was applied and, if not, justify why such corrections were unnecessary.

Thank you for pointing out these oversights. Yes, the p-values associated with multiple hypothesis tests are Bonferroni-corrected. We have updated the detailed information for all the tests that are included in the manuscript (the figure legends of Figure 3-6).

- Reproducibility

It is not clear if the authors provide access to the code or not. To be precise, the proposed architecture has been already published (ref 18). The GitHub provides access to training and test for the DL model. However, there is no script to replicate the analysis presented in the paper, which seems odd. In the attached files the authors say that they will make the SW open source, but it's not clear what SW they refer to.

Sorry for the misunderstanding. Precisely speaking, in this study, we developed a tool for identifying functional non-coding germline variants associated with prostate cancer risk. The tool consists of two components: (1) a deep learning enhancer model, previously published (but not delivered) and located in the 'model' folder, and (2) a novel identification step, which applies a sliding window strategy to the deep learning model to identify functional SNPs (eSNPs) based on the essential window number criterion ($EWN \geq 5$). This identification step is implemented in the 'scoreSNP' folder, listing eSNPs that impact enhancer activity by either gain or loss. We have rephrased the statement as “*The tool developed in this study for identifying eSNPs has been deposited on GitHub (<https://github.com/hannenhalli-lab/ProSNP-DL>).*” at line 505-506 on page 20.

= Minor comments

- Definition of “DL score”: The term "DL score" appears multiple times in the paper, likely referring to "Deep Learning score", but it is never explicitly defined. The authors should provide a clear definition of this term upon first mention to avoid ambiguity.

Thanks for pointing it out. We have rephrased it as “model-predicted score” (Legend of Figure 2 and page 22, line 567-568).

- Clarification of Figures 3C, 3D, and S1C: The method used to generate Figures 3C, 3D, and S1C is unclear. The authors should elaborate on the methodology in the Methods section to ensure reproducibility. Additionally, would it be possible to apply a statistical test to these figures and report a p-value to support the findings quantitative

We regret the lack of clarity. In these three figures, we demonstrated that the AA-dominant eSNPs are more likely to have stronger prostate cancer GWAS signal (the z-

score normalized GWAS signal was provided by the original study (Schumacher, F.R. *et al.*), as compared to the AA-dominant non-eSNPs (Figure 3C), given the cutoff of the GWAS signal (x-axis). In contrast, this is not the case for EA-dominant eSNPs (Figure 3D). The enrichment (y-axis) was measured as the ratio of (i) the fraction of eSNPs with PrCa GWAS signal z-score greater than the signal cutoff (x-axis) to (ii) the fraction of non-eSNPs with PrCa GWAS signal z score greater than the cutoff (x-axis). The Figure S1C directly compares the gained and the lost eSNPs with AA-dominant alleles against gained and lost eSNPs with EA-dominant alleles in terms of PrCa GWAS signals. So overall, these results support an increased enrichment pattern of both gained and lost AA-dominant eSNPs in PrCa GWAS signal, and this enrichment becomes stronger as the GWAS signal cutoff (x-axis) becomes more stringent (Figure 3C and Figure S1C); however, this is not true for EA-dominant eSNPs (Figure 3D). Therefore, the analysis in this part does not involve any methodology per se, but counting the SNPs with strong PrCa GWAS signals and taking the ratio (eSNPs/non-eSNPs) for evaluating the enrichment level. As we need to compare AA-dominant versus EA-dominant for the statistical test, we add Bonferroni-corrected p-values based on Fisher's exact test on Figure S1C as suggested. We have reworded this part in the legend of Figure 3 and added a clarification statement on Page 8, line 183-187 to make it more explicit.

Reviewer #5 (Remarks to the Author):

Reviewer #6 (Remarks to the Author):

Non-coding genetic variants underlying higher prostate cancer risk in men of African ancestry

In this paper the authors employed a sequence-based deep learning model of prostate regulatory enhancers and identified ~2,000 SNPs with higher alternate allele frequency in men of African ancestry that potentially affect enhancer function associated with prostate cancer susceptibility. They explored potential mechanisms associated with the identified enhancer SNPs (eSNPs) that may influence PrCa development and found that the eSNPs could potentially disrupt binding of known prostate transcription factors. They also identified eSNPs that can be combined into a polygenic risk score to add value to current GWAS-based risk variants in assessing PrCa risk in independent cohorts.

Whilst the study is potentially of interest it appears to lack comprehensive training sets to inform 'a deep learning model of enhancer activity' and lacks robust validation using H3K27ac ChIP-seq data from different prostate cell lines. In addition, there are no functional validation experiments to support the proposed effects of eSNPs on their gene targets.

Major criticisms:

1. In developing the machine learning algorithm, the authors only use the one cell line (LNCaP) H3K27ac ChIP-seq data for both training and testing for prostate enhancer prediction. This cell line is not from a normal prostate cell but from an androgen-dependent human prostatic carcinoma cell line originally isolated from a lymph node metastatic lesion and does not represent the diversity of enhancer usage in prostate cells. Expanding the training set to include multiple prostate cell lines would improve the robustness of the model to predict disruption of enhancer function.

See response to comment-2.

2. Similarly, validating that the tool in other prostate cell lines would be more robust than validating in the same cell line. This is feasible given the wide availability of H3K27ac data sets across different prostate cancer cell lines including both androgen-dependent and independent cells.

Thanks for the helpful suggestions. This comment overlapped with the 4th comment of Reviewer 2. Please refer to our detailed response to that comment above. Here we summarize the response. To evaluate the robustness of our model, we trained separate models on VCaP, normal primary prostate tissues, and prostate integrative enhancers—an expanded training set derived from merged enhancers across prostate cell lines and primary tissues. We then evaluated these models on LNCaP cells and vice versa. All models achieve high accuracy, with particularly strong performance when testing LNCaP models on other prostate cell lines and tissues (Response Figure 1AB above). Additionally, the delta scores of 2,000 eSNPs (gained and lost) derived from different enhancer models also correlate well (Response Figure 1C-E above). Specifically, the spearman correlation of delta scores derived from prostate integrative enhancer models and LNCaP models is 0.68 (Response Figure 1E above), further suggesting the robustness of our enhancer model.

3. Critically, the study is lacking functional validation experiments to confirm the effects of identified eSNPs on their proposed gene targets. This could be addressed for example using CRISPR at candidate eSNP loci in LNCaP cells or a 3C investigation at candidate eSNP loci to explore disruption of enhancer promoter loops.

We agree with the reviewer on the importance of functional validations. In fact, we did validate the functional effects of eSNPs in two ways. First, focusing on eSNP loci that differ between LNCaP and MDA Pca 2B cell lines, we have shown an overall differential FOXA1 binding in the two cell lines, at both gained and lost eSNP sites, consistent with the model prediction (Fig 5D). Second, focusing on a specific eSNP in the NDRG1 gene locus, we have shown a higher FOXA1 binding in LNCAP (consistent with model prediction) as well as correspondingly higher expression of NDRG1; higher FOXA1 binding at eSNP and higher NDRG1 expression is consistent with a 3D contact between the eSNP locus and the NDRG1 promoter (Figure 5). We believe that these results indeed functionally validate the model predictions.

However, going beyond NDRG1 locus, to further assess the overall impact of eSNPs on the expression of putative target genes, we compared gene expression differences between the MDA and LNCaP cell lines using expression data from DepMap, focusing on genes that are in 3D contact (via HiChIP loops) with either gained or lost eSNPs. Consistent with expectation, the MDA cell line—which harbors a greater number of alternate alleles at gained eSNPs compared to LNCaP—exhibited higher gene expression at these sites. Conversely, at lost eSNPs, the MDA cell line showed reduced expression relative to LNCaP (Response Figure 2). In short, we experimentally validate the effects of eSNPs, not only on differential FOXA1 binding, but also on the differential expression of putative target genes. We have added this part at line 343-351 Page 15.

Response Figure 2. putative target (linked by HiChIP loops) gene expression change (MDA - LNCaP) of the genes in 3D contact with gained and lost activity eSNP sites where MDA PCa 2B (dubbed MDA) genotype has a greater number of alternate alleles. * refers to Bonferroni-corrected Wilcoxon P-values < 0.05.

Other important analyses:

4. Please provide statistics for Fig 3E. Fraction of genes >1.5 HR for gained eSNPs is very close to the value for all LNCaP enhancers which diminishes any specific effects of gained eSNPs on HR. Also, the related figure Supp 1D shows a similar fraction of genes with >1.5 HR for EA lost and gained SNPs. This seems contradictory to the finding in Figure 3D which shows EA lost and gained eSNPs are not associated with prostate cancer GWAS traits.

We regret the lack of clarity. In the original plot of Fig 3E and Fig S2D, we only consider the universe of genes (denominator) with significant HRs ($p\text{-value} \leq 0.05$) (Response Fig 3AB). We have now considered all enhancer-associated genes in each category and re-estimated the fraction and p-values to the two plots (Response Fig 3CD, i.e. Fig 3E and Fig S2D). The overall trend remained unchanged. However, as before, based

on Fisher test, the fraction of genes with $HR > 1.5$ for AA-dominant gained eSNPs genes are clearly greater but not significantly so than those of the all LNCaP enhancers. Considering that this lack of significance could be due to small dataset, we estimated a nominal p-value based on a bootstrap approach as follows: we randomly sampled 1,097 genes from the 9,840 genes associated with all LNCaP enhancers 100 times, and for each sample, calculated the fraction of genes with a high hazard ratio ($HR > 1.5$, $p \leq 0.05$). Overall, the observed fraction of high-HR genes among gained eSNPs (0.0273) falls within the top 0.0189 percentile of the distribution of these 100 sample fractions, yielding nominal p-value of 0.0189. We have now added this part to the Method (page 23). With regards to Fig 3D and Fig S1D, to clarify, the enrichment of GWAS signal is seen only for AA- dominant eSNPs (the y-axis in Fig 3C > 1), and in contrast, the EA-dominant eSNPs do not show this trend (the y-axis in Fig 3D mostly less than 1, consistent with Fig S2D).

Response Figure 3. A) Fraction of genes with HR > 1.5 (Bonferroni-corrected P-value ≤ 0.05) associated with AA-dominant gained activity eSNPs, lost activity eSNPs, and all LNCaP enhancers. Only genes with significant HRs (Bonferroni-corrected P-value ≤ 0.05) were included in this plot B) Fraction of genes with HR > 1.5 (Bonferroni-corrected P-value ≤ 0.05) associated with EA-dominant gained activity eSNPs, lost

activity eSNPs, and all LNCaP enhancers. Only genes with significant HRs (Bonferroni-corrected P-value ≤ 0.05) were included in this plot C) In the three categories -- AA-dominant gained activity eSNPs, lost activity eSNPs, and all LNCaP enhancers, the figures shows the fraction of genes with HR > 1.5 (Bonferroni-corrected P-value ≤ 0.05). Each pair of categories was compared using Fisher's exact test. n.s. refers to non-significant based on Fisher's exact test. * refers to significance using bootstrap-based approach: p-value = 0.0189 (Methods). D) In the three categories -- EA-dominant gained activity eSNPs, lost activity eSNPs, and all LNCaP enhancers, the figure shows the fraction of genes with HR > 1.5 (Bonferroni-corrected P-value ≤ 0.05). Each pair of categories was compared using Fisher's exact test. n.s. refers to non-significant based on Fisher's exact test.

5. For the GO analysis the methods state on line 558 - 'To probe the potential functional roles of gained and lost eSNPs, we tested for functional enrichment among genes near the eSNPs loci using the online Genomic Regions Enrichment of Annotations Tool (GREAT23) version 3.0.0 using two-nearest-genes association rule... The whole genome region was used as the background.' –Whole genome is not the correct background. The authors need to control for genomic biases in location of eSNPs or SNPs, e.g. using genes next to EA dominant eSNPs or non eSNPs as background.

Thanks for the suggestions. We have now repeated the analysis using the genome-wide random non-eSNPs as the background (Response Figure 3 and Fig 4AB). We observed a similar enrichment pattern when using 500K random non eSNPs as before based on the whole genome as the background, where gained eSNPs are linked to the regulation of the immune system and telomere length, while lost eSNPs are associated with development/differentiation. We have made corresponding changes in Figure 4AB and the method section at line 612-625 on page 23 accordingly.

Response Figure 3. A) Biological processes associated with gained activity eSNPs using the GREAT tool. B) Biological processes associated with lost eSNPs using the GREAT tool.

6. In Figure 4D, why do genes proximal to gained eSNPs exhibit significantly higher expression in fetal prostate if they are regulating immune function? All of the pathways highlighted for gained eSNPs are all essential in adult function too.

Based on our analysis, the genes proximal to gained eSNPs are in fact largely involved in suppression of immune system (Fig 4A and C), which is an essential biological

process in fetal development (Yu, J. *et al.* 2024, Aghaeepour, N. *et al.* 2017). Therefore, it doesn't seem surprising that these genes are highly expressed in the fetal prostate compared to the adult prostate. We have clarified this interpretation at line 244-248 on page 11: *“It is worth noting that, the genes proximal to gained eSNPs are largely involved in suppression of immune system (Fig 4A and C), which is an essential feature of fetal development^{30,31}. Therefore, it does not seem surprising that these genes are highly expressed in the fetal prostate compared to the adult prostate.”*.

7. Line 232 – ‘Taken together, these results strongly suggest that by activating oncogenic processes and disrupting tumour suppression mechanisms, the gained and lost activity eSNPS collaboratively promote carcinogenesis – This is not supported by the GO analysis for gained eSNPS which mainly includes immune and lipid metabolism terms. Similarly for lost eSNPs, there is a large conceptual gap between lost SNPs being near genes that regulate differentiation development and lost SNPs being involved in disrupting ‘tumour suppression mechanisms’. Also see point about line 263 below. Please modify the language of the conclusions over to reflect the actual data.

We appreciate this comment. Besides making appropriate language changes, we have performed the following additional analysis. To further investigate the relevance of eSNP-associated genes to tumor progression—specifically their pro- or anti-tumorigenic potential—we evaluated whether genes linked to gained eSNPs, particularly those enriched in immune response, regulation of immune response, and regulation of telomere maintenance Gene Ontology (GO) terms, are associated with poorer patient survival, and whether genes linked to lost eSNPs, enriched in developmental and differentiation-related GO terms, correlate with improved survival outcomes. Notably, genes within the enriched GO terms associated with gained eSNPs consistently exhibited elevated hazard ratios ($HR > 1$), indicative of pro-tumorigenic role, whereas those associated with lost eSNPs consistently showed reduced hazard ratios ($HR < 1$) (Response Figure 4), suggestive of tumor-suppressive functions. Together, these findings support our interpretation that gained and lost eSNPs may collaboratively promote carcinogenesis through complementary mechanisms—by activating oncogenic pathways and impairing tumor suppressive functions. We have modified this section accordingly (Page 11, line 250-261).

Response Figure 4. Hazard Ratios of gene targets of gained and lost eSNPs corresponding to the enriched GO terms. Only genes with significant HRs (Bonferroni-corrected P-value ≤ 0.05) were included in the plot. Along x-axis, purple dots are GO terms associated with differentiation/development, light blue dots are associated with regulation of the immune system, dark blue dots are associated with immune response, and red dots are GO terms associated with telomere length regulation.

8. Line 248 – ‘Likely to be mediated’ is also too strong as only the author refer to only one reference (ref 17) that has shown that one SNP acts via this mechanism but did not provide functional evidence themselves.

We have toned down the language and added further references more directly supporting the statement on page 13, line 276-278.

9. Line 263 – To the best of my understanding the CancerSEA signatures all pertain to functional cancer states. The CancerSEA publication mentions nothing about some states being ‘pro-tumor’ vs ‘anti-tumour’, rather, the authors have just ascribed these descriptions to some of the states without explaining how or why. The same occurs in lines 389-392 in the discussion.

Thanks again for pointing out this oversight. We did similar analysis as above (the 7th comment, GO term) to investigate the pro-tumor and anti-tumor potential of eSNP genes in the enriched CancerSEA signatures. Consistent with our hypothesis, the target genes of gained eSNPs in Stemness, Metastasis, and Hypoxia CancerSEA signatures, are associated with worse survival (HR > 1), suggestive of pro-tumorigenic potential. In contrast, genes for lost eSNPs belonging to Differentiation and Apoptosis signatures are associated with better prognosis (HR < 1), indicative of tumor-suppressive functions. We have added this additional analysis at line 291-304 on page 13.

Response Figure 5. Hazard Ratios of gene targets of gained and lost eSNPs belonging

to enriched CancerSEA signatures. Only genes with significant HRs (Bonferroni-corrected P-value ≤ 0.05) were included in the plot.

10. Figure 5D and 5E – Authors should provide details on how they performed FOXA1 ChIP-seq and analyses; it is not mentioned in the results or methods and therefore it is not possible to determine if the claims of loss/gain of peaks can be supported by the data presented. Were the ChIPs performed in triplicates, how is the ChIP-seq data quantitative, ie were spike ins added or normalization performed to allow comparison of signal intensities across samples? Did the ChIPs work effectively in both cell-lines to permit a quantitative comparison of FOXA1 binding efficiencies? Further to this, the candidate screenshot in 5E shows a very zoomed in area of the peak. It is important to provide QC data to determine that the FOXA1 signal genome-wide is limited to FOXA1 binding sites and is absent elsewhere in the genome as a testament to the quality of the data.

We agree with the reviewer that based on the data currently presented it is difficult to determine if the quality of the FOXA1 ChIP-seq experiments is sufficient to conclude differences in FOXA1 binding. Per current ENCODE consortium guidelines for transcription factor ChIP-seq, we performed two biological replicates with corresponding input control. Library complexity as determined by non-redundant fraction (NRF) and PCR bottlenecking coefficients 1 and 2 (PBC1, PBC2) passed recommended values. Additional quality assessment metrics such as fraction of reads in peaks (FRiP score) are provided in Table below. Notably the most enriched motif in significant peaks is FOXA1 with p-value an order of magnitude greater than next enriched motif, consistent with specificity of ChIP-seq for FOXA1.

LNCaP Replicate1:

Rank	Motif	P-value	log P-pvalue	% of Targets	% of Background	STD(Bg STD)	Best Match/Details
1		1e-11314	-2.605e+04	59.77%	16.80%	206.0bp (123.6bp)	FOXA1 (Forkhead)/LNCAP-FOXA1-ChIP-Seq(GSE27824)/Homer(0.980) More Information Similar Motifs Found
2		1e-1275	-2.938e+03	30.21%	17.02%	226.4bp (119.3bp)	Hoxd13(Homeobox)/ChickenMSG-Hoxd13.Flag-ChIP-Seq(GSE86088)/Homer(0.968) More Information Similar Motifs Found
3		1e-970	-2.235e+03	38.91%	25.97%	207.3bp (122.5bp)	NFIX/MA0671.1/Jaspar(0.972) More Information Similar Motifs Found
4		1e-306	-7.049e+02	9.00%	5.15%	220.2bp (115.2bp)	ELF3(ETS)/PDAC-ELF3-ChIP-Seq(GSE64557)/Homer(0.987) More Information Similar Motifs Found
5		1e-293	-6.767e+02	4.10%	1.72%	214.0bp (115.0bp)	GATA(Zf),IR3/iTreg-Gata3-ChIP-Seq(GSE20898)/Homer(0.957) More Information Similar Motifs Found

LNCaP Replicate2:

Rank	Motif	P-value	log P-pvalue	% of Targets	% of Background	STD(Bg STD)	Best Match/Details
1		1e-13058	-3.007e+04	59.68%	16.60%	194.8bp (113.2bp)	FOXA1 (Forkhead)/LNCAP-FOXA1-ChIP-Seq(GSE27824)/Homer(0.982) More Information Similar Motifs Found
2		1e-1289	-2.970e+03	27.49%	15.51%	211.7bp (111.3bp)	Hoxa13(Homeobox)/ChickenMSG-Hoxa13_Flag-ChIP-Seq(GSE86088)/Homer(0.900) More Information Similar Motifs Found
3		1e-1082	-2.492e+03	32.69%	20.72%	200.3bp (113.3bp)	NFIX/MA0671.1/Jaspar(0.976) More Information Similar Motifs Found
4		1e-345	-7.951e+02	4.16%	1.74%	215.7bp (103.7bp)	GATA(Zf).IR4/iTreg-Gata3-ChIP-Seq(GSE20898)/Homer(0.941) More Information Similar Motifs Found
5		1e-338	-7.800e+02	12.86%	8.26%	209.0bp (108.6bp)	EHF(ETS)/LoVo-EHF-ChIP-Seq(GSE49402)/Homer(0.975) More Information Similar Motifs Found
6		1e-264	-6.084e+02	3.10%	1.28%	204.9bp (106.6bp)	GATA(Zf).IR3/iTreg-Gata3-ChIP-Seq(GSE20898)/Homer(0.979) More Information Similar Motifs Found

MDA_PCa2b Replicate 1:

Rank	Motif	P-value	log P-pvalue	% of Targets	% of Background	STD(Bg STD)	Best Match/Details
1		1e-8987	-2.069e+04	59.99%	15.12%	170.3bp (115.5bp)	FOXA1 (Forkhead)/LNCAP-FOXA1-ChIP-Seq(GSE27824)/Homer(0.980) More Information Similar Motifs Found
2		1e-1099	-2.532e+03	36.78%	21.11%	186.9bp (117.0bp)	HOXB13(Homeobox)/ProstateTumor-HOXB13-ChIP-Seq(GSE56288)/Homer(0.934) More Information Similar Motifs Found
3		1e-742	-1.709e+03	40.41%	26.81%	181.5bp (119.1bp)	ZSCAN29/MA1602.1/Jaspar(0.737) More Information Similar Motifs Found
4		1e-485	-1.118e+03	9.62%	4.11%	182.1bp (113.7bp)	NF1:FOXA1(CTF,Forkhead)/LNCAP-FOXA1-ChIP-Seq(GSE27824)/Homer(0.829) More Information Similar Motifs Found
5		1e-349	-8.038e+02	5.70%	2.17%	183.5bp (108.3bp)	GATA(Zf).IR3/iTreg-Gata3-ChIP-Seq(GSE20898)/Homer(0.980) More Information Similar Motifs Found

MDA_PCa2b Replicate 2:

Rank	Motif	P-value	log P-pvalue	% of Targets	% of Background	STD(Bg STD)	Best Match/Details
1		1e-13815	-3.181e+04	64.21%	21.01%	199.8bp (113.9bp)	FOXA1 (Forkhead)/LNCAP-FOXA1-ChIP-Seq(GSE27824)/Homer(0.991) More Information Similar Motifs Found
2		1e-1255	-2.891e+03	23.63%	13.25%	216.8bp (110.8bp)	PH0068.1_Hoxc13/Jaspar(0.901) More Information Similar Motifs Found
3		1e-622	-1.433e+03	17.71%	11.05%	218.6bp (112.2bp)	POL012.1_TATA-Box/Jaspar(0.767) More Information Similar Motifs Found
4		1e-608	-1.401e+03	30.02%	21.65%	223.1bp (110.0bp)	GATA6/MA1104.2/Jaspar(0.976) More Information Similar Motifs Found
5		1e-528	-1.216e+03	24.65%	17.42%	209.9bp (113.9bp)	NF1-halfsite(CTF)/LNCaP-NF1-ChIP-Seq(Unpublished)/Homer(0.977) More Information Similar Motifs Found

For Figure 5D, we used bamCoverage from deeptools/3.5.1 to normalize for read depth using reads per genomic content (RPGC) with a bin size of 25 bp and effective genome size of 2.7 billion. This description has been added to the section of Data processing in Methods. We then took the average of the normalized signals of duplicates of each cell line for comparison of signal intensities across the two cell lines. We also provided a zoomed out version of Figure 5E to demonstrate the visually distinct peak overlapping with the SNP.

Response Figure 6. A zoomed out version of Fig 5E. Read coverage of FOXA1 at rs10095018 in LNCaP and MDA cell lines, where the alternate allele C disrupts the binding of FOXA1 according to the consensus motif of FOXA.

In addition, here are some QC metrics for those ChIP experiments. Note the top scoring peak is forkhead with very significant p-value.

SampleName	Unique_Mapped	Dup_Rate	NRF	PBC1	PBC2	FRIP	Peaks	Peaks>10	Motif	p_value
KF3_incap_foxa1_S2	86.18	0.07	0.915	0.928	14.08	0.16144	72502	27279	FOXA1	1e-11314
KF6_incap_foxa1_kf_S7	79.15	0.09	0.912	0.914	11.735	0.18293	83948	34488	FOXA1	1e-13058
KF3_incap_input_S1	54.78	0.06	0.933	0.94	16.854	NA	NA	NA	NA	NA
KF6_incap_input_S5	98.15	0.07	0.929	0.934	15.274	NA	NA	NA	NA	NA
KF3_mda_foxa1_S4	52.48	0.06	0.932	0.937	15.876	0.11506	52048	21465	FOXA1	1e-8987
KF6_mda_foxa1_kf_S10	57.27	0.09	0.91	0.911	11.245	0.23992	98662	42886	FOXA1	1e-13815
KF3_mda_input_S3	51.7	0.06	0.929	0.937	16.023	NA	NA	NA	NA	NA
KF6_mda_input_S8	54.49	0.07	0.932	0.934	15.281	NA	NA	NA	NA	NA

QC metrics description

- Unique_Mapped Uniquely mapped reads (millions)
- Dup_Rate Duplication Rate
- NRF Non redundant fraction (> 0.9 is ideal according to ENCODE)
- PBC1 PCR bottlenecking coefficient 1 (> 0.9 is none according to ENCODE standards)
- PBC2 PCR bottlenecking coefficient 2 (> 3 is none according to ENCODE standards)
- FRIP Fraction of reads in peaks

Peaks	Number of peaks called by MACS
Peaks > 10	Number of peaks called by MACS with fold change > 10
Motif	Top scoring motif by Homer
p_value	p-value of top scoring hit by Homer

We have also added this part to the section of “FOXA1 ChIP-seq experiments in two PrCa cell lines” in Supplementary information section II.

Reviewer #6 (Remarks on code availability):

I don't have the expertise to review the code.

References

- Horoszewicz, J.S. *et al.* LNCaP model of human prostatic carcinoma. *Cancer Res* **43**, 1809-18 (1983).
- Navone, N.M. *et al.* Establishment of two human prostate cancer cell lines derived from a single bone metastasis. *Clin Cancer Res* **3**, 2493-500 (1997)
- Schumacher, F.R. *et al.* Association analyses of more than 140,000 men identify 63 new prostate cancer susceptibility loci. *Nat Genet* **50**, 928-936 (2018).
- Yu, J. *et al.* Progesterone-driven B7-H4 contributes to onco-fetal immune tolerance. *Cell* **187**, 4713-4732 e19 (2024).
- Aghaeepour, N. *et al.* An immune clock of human pregnancy. *Sci Immunol* **2**(2017).
- Pomerantz *et al.* Prostate cancer reactivates developmental epigenomic programs during metastatic progression. *Nat Genet* **52**, 790-799 (2020).

REVIEWERS' COMMENTS

Reviewer #1 (Remarks to the Author):

I believe the results of this revised manuscript remain noteworthy. The authors identified non-coding regulatory polymorphisms associated with PrCa in African American (AA) men. Men of AA ancestry shared a disproportionate burden of PrCa morbidity and mortality and are underrepresented in PrCa genomics research. They identified enhancer SNPs (eSNPs) which are associated with apoptosis and immunosuppression which have significant roles in cancer development and progression.

I concur with the authors that the role of non-coding regulatory polymorphisms in PrCa has not fully investigated. I also believe this report could add a novel aspect of including eSNPs in the construction of ancestry-dependent polygenic risk scores for PrCa.

This revised work is more comprehensive. In my view, the authors have taken great care to sufficiently address the concerns of all reviewers. The manuscript would add to the existing literature in the area of PrCa genomics. The methodology is sound and meets the expected standards in the PrCa genomics field. I believe this revised manuscript also provides enough detail in the methods to be reproducible.

Overall, I do not detect any unresolved and/or newly recognized major flaws in the revised manuscript's methodology, data analysis, interpretations and conclusions. I do not believe that this manuscript requires further revision, and I think it is now suitable for publication.

Reviewer #1 (Remarks on code availability):

N/A

Reviewer #2 (Remarks to the Author):

I am satisfied with the reply of the authors to my comments. Their confirmation that the results closely resemble those of other prostate cancer cell lines and normal primary prostate tissues, and the inclusion of this information in the manuscript, render the investigation more robust.

Reviewer #3 (Remarks to the Author):

I co-reviewed this manuscript with one of the reviewers who provided the listed reports. This is part of the Nature Communications initiative to facilitate training in peer review

and to provide appropriate recognition for Early Career Researchers who co-review manuscripts.

Reviewer #4 (Remarks to the Author):

The reviewers have answered our concerns and improved the current version of the manuscript.

Reviewer #5 (Remarks to the Author):

Reviewer #5 (Remarks on code availability):

The code of the model is present but not the full code to reproduce the analysis.

Reviewer #7 (Remarks to the Author): Expert in prostate cancer epigenetics and epigenomics, gene regulation; replaces Reviewer #6

In this manuscript Li et al. have aimed to interrogate single-nucleotide polymorphisms (SNPs) underlying increased prostate cancer risk and mortality in men of African-American ancestry. The authors approached this question by employing a sequence-based deep learning model of regulatory H3K27ac enhancers associated with ~2000 eSNPs that are present at a higher alternate frequency in AA men and potentially associated with PC susceptibility. The authors noted that eSNPs tend to disrupt the binding of FOX, AR and HOX families.

Overall, the manuscript is interesting, comprehensive and is systematic in investigating the role of nucleotide polymorphisms. The authors have also addressed a majority of the concerns of the previous reviewers.

Comments:

Line 26: The authors mention the eSNP disrupt the binding of known prostate TFs including FOXA1, AR and HOXB13. Also, Line 274 of the manuscript mentions the same. In Line 277, the disruption is described as (gain or loss) of TF binding. The term

“disrupt” is confusing, as it would align more with loss of function of TF binding and tends to deemphasize the gain of TF activity at H3K27ac enriched eSNPs. A revision of this term is suggested.

We've changed the word “disrupt” to “modulate” in these two sections.

From the mechanistic/validation perspective the manuscript has some weaknesses as also pointed out by the earlier reviewer. For e.g in Fig 5E, the eSNP at rs10095018 reduces but not completely disrupts FOXA1 binding in MDA-PCa2B cell line (AA ancestry) compared to LNCaP cell line (EA ancestry). There is no experimental data provided for AR or HOXB13 TF binding at these H3K27ac eSNPs. Likewise, whether AR and HOXB13 binding is affected due to disrupted FOXA1 at eSNP is not demonstrated. Lacking is the important ChIP-seq data for H3K27ac/FOXA1/AR/HOXB13 at this site in these PC two cell lines under the same experimental conditions, or in primary prostate cancer tissues of EA and AA origins, to demonstrate that this indeed there is overlap of eSNP associated with the enhancer activity and TF binding. Intriguingly, the percentage of potential FOXA1 targets is comparable in LNCaP (59.68% and 59.77%) and MDA-PCa2b (59.99% and 64.21%). Therefore, it is not clear whether FOXA1 binding alone can explain the dominant activity of eSNPs in AA versus EA men. This aspect requires revision or further clarification by authors to support their claims.

The reason we chose FOXA1 as a candidate TF whose binding is potentially impacted by the eSNPs is because 1) the eSNPs affecting FOX family (especially FOXA1) not only exhibited large effects on enhancer activity (average EWN ≥ 7) but also had the highest number of potential target genes (Fig. 5AB). Specifically, the average number of target genes of the FOX family is ~ 2 -fold of those of the HOX family, and at least 3-fold of those of AR (Fig. 5AB). 2) FOXA1 functions as a pioneer factor in prostate cancer, capable of extensively reprogramming the AR cistrome in collaboration with HOXB13 (Teng, M. et. al., 2021). Therefore, we reasoned that the modulation of FOXA1 binding is most likely to affect enhancer activity and the binding of many other essential TFs including HOXB13 and AR. Notably, when further assessing the overall impact of eSNPs on the expression of all the putative target genes regardless of whether they are regulated by FOXA1, we compared gene expression differences between the MDA and LNCaP cell lines using expression data from DepMap, focusing on genes that are in 3D contact (via HiChIP loops) with either gained or lost eSNPs. Consistent with expectation, the MDA cell line—which harbors a greater number of alternate alleles at gained eSNPs compared to LNCaP—exhibited higher gene expression at these sites. Conversely, at lost eSNPs, the MDA cell line showed reduced expression relative to LNCaP (Fig. S6). While we agree that a more complete characterization of rs10095018 will require assessment of binding of other TFs including AR and HOX, our computational analysis combined with FOXA1 ChIP-seq does provide compelling evidence for the role of FOXA1 binding modulation at this SNP.

Reviewer #7 (Remarks on code availability):

Not tested.

References:

Teng, M., Zhou, S., Cai, C., Lupien, M. & He, H.H. Pioneer of prostate cancer: past, present and the future of FOXA1. *Protein Cell* **12**, 29-38 (2021).